# Soil Type Influences Novel “Milpa” Isolates of *Trichoderma virens* and *Aspergillus tubingensis* That Promote Solubilization, Mineralization, and Phytoabsorption of Phosphorus in *Capsicum annuum* L.

**DOI:** 10.3390/jof8111178

**Published:** 2022-11-08

**Authors:** Dorcas Zúñiga-Silgado, Ayixon Sánchez-Reyes, María Laura Ortiz-Hernández, Miranda Otero, Edgar Balcázar-López, Susana Valencia-Díaz, Mario Serrano, Jeffrey J. Coleman, Luis Sarmiento-López, Luz E. De-Bashan, Jorge Luis Folch-Mallol

**Affiliations:** 1Centro de Investigación en Biotecnología, Universidad Autónoma del Estado de Morelos, Av. Universidad No. 1001, Col. Chamilpa, Cuernavaca 62209, Mexico; 2Investigador por México, CONACyT, Instituto de Biotecnología, Universidad Nacional Autónoma de México, Ave. Universidad 2001, Col. Chamilpa, Cuernavaca 62210, Mexico; 3Misión Sustentabilidad México A.C., Priv Laureles 6, Col. Chamilpa., Cuernavaca 62210, Mexico; 4Department of Entomology and Plant Pathology, Auburn University, Rouse Life Sciences Building, Auburn, AL 36849, USA; 5Departamento de Farmacobiología, Centro Universitario de Ciencias Exactas e Ingeniería, Universidad de Guadalajara, Blvd. M. García Barragán #1451, Guadalajara 44430, Mexico; 6Centro de Ciencias Genómicas, Universidad Nacional Autónoma de México, Ave. Universidad 2001, Col. Chamilpa, Cuernavaca 62210, Mexico; 7Departamento de Biotecnología Agrícola, Centro Interdisciplinario de Investigación para el Desarrollo Integral Regional Unidad Sinaloa-Instituto Politécnico Nacional, Guasave 81288, Mexico; 8Bashan Institute of Sciences, 1730 Post Oak Ct, Auburn, AL 36830, USA; 9Environmental Microbiology Group, Northwestern Center for Biological Research (CIBNOR), Av. IPN 195, La Paz 23096, Mexico

**Keywords:** milpa fungal diversity, phosphorus bioavailability, plant-growth promotion, Vertisol, Andisol, germination speed, biomass yield

## Abstract

The *Capsicum* genus has significant economic importance since it is cultivated and consumed worldwide for its flavor and pungent properties. In 2021, Mexico produced 3.3 billion tons on 45,000 hectares which yielded USD 2 billion in exports to the USA, Canada, Japan, etc. Soil type has a dramatic effect on phosphorus (P) availability for plants due to its ion retention. In a previous study, novel fungal isolates were shown to solubilize and mineralize P in different kinds of soils with different P retention capacities. The aim of this work was to study the effects of the mineralogy of different kinds of “milpa” soils on the germination, biomass production, and P absorption of chili plants (*Capsicum annuum*). The germination percentage, the germination speed index, and the mean germination time were significantly increased in the plants treated with dual inoculation. Foliar phosphorus, growth variables, and plant biomass of chili plants grown in a greenhouse were enhanced in different soil types and with different inocula. Correlation studies suggested that the most significant performance in the foliar P concentration and in the growth response of plants was achieved in Vertisol with dual inoculation of 7 × 10^6^ mL^−1^ spores per chili plant, suggesting this would be an appropriate approach to enhance chili cultivation depending on the soil type. This study stresses the importance of careful analysis of the effect of the soil type in the plant–microbe interactions.

## 1. Introduction

The *Capsicum* genus comprises at least 35 species and belongs to the Solanaceae family. Phylogenetic information suggests that chili originated from a region in the Northwestern Andes, from which it spread to the Amazon basin, passed to the Caribbean Islands, and then passed to Mexico [1]. However, Kraft et al. place the origin of domesticated species in central-east Mexico using archaeological evidence, paleobiolinguistics, and genetic data [2]. Mexico produces approximately 3 thousand million tons of chili annually, valued at nearly USD 2.3 billion [3,4].

For the cultivation of crops, phosphorus (P) is considered the macronutrient with the lowest mobility and availability for plants in most tropical soils [5]; thus, it is one of the most critical elements limiting plant growth. The low mobility of bioavailable phosphate ions (H_2_PO_4_^−^ and HPO_4_^=^) is due to their retention by the colloidal mineral constituents of the soil [6], and therefore only a small proportion of the available ions is present in the soil [7]. The quantity and quality of nutrients accumulated in tropical soils, as well as the decomposition rate of organic matter, are important characteristics for the global functioning of any edaphic ecosystem for agricultural production purposes. Sequential extraction of P is widely used to obtain various fractions of inorganic P (Pi) and organic P (Po) that differ in their availability for microorganisms and plants [8]. However, data on P fractions in Andisol and Vertisol soils of Mexico have not been reported. All along the country, an agroecological pre-Hispanic system called “milpa” is used. It consists of growing several plant species together in the same area, in which corn, beans, squash, and chili are always present (other plant species may be included in different parts of the country). There are scarce reports of fungal communities in milpas, and most of them refer to arbuscular mycorrhizae or are only available in Spanish [9]. 

The study of the different forms of P in these soils and how they are affected could increase our understanding of the P cycle and agronomic practices in native soils. In this regard, the application of rock phosphate (RP) has been extensively studied as a method to satisfy the phosphoric requirements of crops [10,11]. However, the amount of available P released to the soil from the direct application of RP is too low to suffice the demand of the crops, limiting its widespread use [12,13]. Faced with this situation, the need arises to implement low-cost technological alternatives to improve the acquisition of P by plants. The use of rhizospheric microorganisms with the capacity to solubilize and mineralize P has proven to be an alternative [14,15]. The rhizospheric microbiota represents a dynamic turnover of nutrients that responds quickly to changes in the soil, such as changes in pH, water retention, and ionic force. In this sense, microorganisms play a key role in the transformation of P in the soil through the excretion of phosphatases and/or organic acids. The *Trichoderma* genus is one of the most studied fungal genera for its plant-growth-promoting activity [16,17]. Its plant-growth-promoting effect is due to several properties such as being a mycoparasite that controls fungal diseases in plants; it triggers the defense mechanisms of the plant via jasmonic and salicylic acids, as well as the ethylene pathway. Finally, it has been shown that the fungi acidify the soil, making insoluble nutrients available to the plant and modifying root architecture. *Aspergillus* spp. have not been studied as extensively as *Trichoderma*, but there are a few reports in which they have been shown to secrete organic acids, thus making nutrients more available to the plant [18]. These fungi are capable of solubilizing phosphorus or potassium, decomposing organic material, or oxidizing sulfur in the soil, thus improving agricultural production by increasing the supply of nutrients [19]. Therefore, their use as biofertilizers can partially replace chemical fertilizers and promote higher growth and yield of plants [16]. Microbial consortia as biofertilizers have been successfully used for various crops of agronomic importance, including chili in Mexico [20]. However, the mechanisms of growth promotion according to the type of soil where they are grown remain mostly unknown. 

The physicochemical condition of the rhizosphere is the main influencer of the proliferation of microbial functional groups, and pH is of key importance in determining the ecosystem and productive success of plants and soil microorganisms [21,22,23]. Among plant species, the optimum soil pH for growth can vary considerably, although most crops grow well at slightly acidic or neutral soils [6,24]. However, when soil pH becomes lower than 5.5, root growth is impaired and plant growth decreases. Soil acidity is correlated with an increase in toxic levels of aluminum (Al), manganese (Mn), iron (Fe), and protons (H^+^), as well as a decrease in the availability of P, calcium (Ca), and magnesium (Mg) [24,25]. Although the solubilization and mineralization capacity of P is associated with a decrease in pH due to the release of organic acids (OAs) or secretion of phytases, the effect of the soil mineralogy on the secretion of these substances in the solubilization and mineralization efficiency of P has been only partially documented [26]. 

This study follows our previous hypothesis that the soil mineralogy of two studied milpas and the attributes of the fungal inoculum will modulate the efficiency of in vitro RP microbial solubilization and phytoabsorption of P. Thus, the objective of this research was to evaluate the effects of soil mineralogy, composition, and concentration of the inoculum of previously described *Trichoderma* and *Aspergillus* species on the germination, P absorption, and growth responses of *C. annuum* L.

## 2. Materials and Methods

### 2.1. Sampling Area

Fungal isolates were collected from two locations in the state of Morelos, Mexico, corresponding to Horizon A (0 to 20 cm) of an Andisol (Tres Marías, Municipality of Huitzilac; 19°02′18″ N 99°15′11″ W; 2803 m above sea level (masl)) and a Vertisol (Cuentepec, Municipality of Temixco; 18°51′36″ N 99°19′29″ W; 1480 masl). The physicochemical composition of the soils is shown in modified Appendix A previously described in [26].

### 2.2. Fungal Biodiversity as Measured Macro- and Microscopically in Different Selective Media

From each of the sampled milpas and microhabitats (rhizosphere, rhizoplane, and root endophytic samples), serial dilutions were performed and seeded in Petri dishes using selective media for phosphorus uptake: OHM medium (for solubilizers) [27] (in g L^−1^): 1.0 NaCl; 0.2 CaCl_2_·2H_2_O; 0.4 MgSO_4_·7H_2_O; 1.0 NH_4_NO_3_; 10 glucose; 3.5 non-acidulated RP (Fosforita 28P, which is composed of 8% Ca_5_(PO_4_)_3_, 38% Ca(OH)_2_, 14% SiO_2_, 3% F, 1% C, 0.50% Al_2_O_3_, 0.40% Fe_2_O_3_, 0.10% Mg(OH)_2_ 0.30% SO_4_, 10% Na_2_O, 0.10% K_2_)_._

For mineralizers, PNM (Sigma, St. Louis, MO, USA) was used (in g L^−1^): 1.0 NaCl; 0.2 CaCl_2_·2H_2_O; 0.4 MgSO_4_·7H_2_O; 1.0 NH_4_NO_3_; 10 glucose, 10 Na^+^ Phytate. The Na^+^ Phytate was added after the medium was autoclaved, when the final temperature was 28 ± 1 °C. Bengal Rose Agar (BRA) and Potato Dextrose Agar (PDA) media were purchased from Sigma (St. Louis, MO, USA). All the media were supplemented with ampicillin (100 μg mL^−1^) and chloramphenicol (30 μg mL^−1^) to avoid bacterial growth. The dishes were incubated at 28 °C until fungal growth was observed. The first selection was carried out in the mineralizing medium (PNM) and the solubilization medium (OHM). From the colonies that grew in these media, three monosporic cultures were performed in PDA for each strain to ensure the purity of the isolate. BRA medium was used for the selection of *Trichoderma* spp. and to differentiate them from *Rhizopus* growth. In all cases, the initial 10^−6^ dilution yielded approximately 300 colonies, and these numbers were used to make the calculations of CFU per gram of soil or root tissue in the case of endophytes. The different isolates were classified according to their macro- and microscopic appearance according to [28].

### 2.3. Selection of P-Solubilizing and -Mineralizing Strains

To evaluate the effect of the medium with different soil types on the solubilization or mineralization capacity of the strains, the diameter of the colonies (colony diameter, CD) was measured as a function of time (3, 6, 9, and 12 days after inoculation). Additionally, the solubilization halo diameter (SHD) and the relative efficiency of RP solubilization (RES) were measured [26].

### 2.4. Biocompatibility among Strains

Thirty-one fungal isolates from the previous work [26] were tested in OHM and PNM for biocompatibility by seeding them in the same dish and observing the presence or absence of a defense barrier between the strains. The 31 fungal morphotypes were grown on PDA medium for 6–8 days (depending on the growth rate), and spore suspensions with 7 × 10^−6^ spores mL^−1^ were prepared. From these suspensions, one hundred microliters of each suspension were sown on opposite sides of the dish for each combination which contained two, three, or four strains in the same Petri dish (Figure 1) and incubated for 5 days at 25 ± 1 °C. The growth rate was also determined in both media.

Strains BMH-0059, BMH-0060, BMH-0061, and BMH-0062 were further studied based on their P mineralization and solubilization capacity, as well as their biocompatibility.

### 2.5. In Vitro Effect of the Isolates on Germination of C. annuum L.

For these experiments, isolates BMH-0059 (*Trichoderma virens*), BMH-0060 (*Aspergillus tubingensis*), BMH-0061 (*Trichoderma* sp.), and BMH-0062 (*Trichoderma pubescens*) were used. To evaluate the effect of the inoculum concentration on the germination of chili seeds, spore suspensions were prepared from the freshly grown strains in PDA collected in CaCl_2_·2H_2_O 0.01 M at three different concentrations, namely 7 × 10^4^ (low), 7 × 10^6^, (medium) and 7 × 10^8^ spores mL^−1^ (high), of each fungal strain; the volumes were adjusted using a Newbauer chamber. Ten chili seeds per Petri dish were inoculated with 5 mL of each spore suspension. In each Petri dish, a cellulose membrane was placed. As a control, non-inoculated Petri dishes received 5 mL of the 0.01 M CaCl_2_·2H_2_O solution. A total of 13 treatments were evaluated (the 4 selected strains and 3 inoculum concentrations plus the dual inoculation of *T. virens* BMH-0059 and *A. tubingensis* BMH-0060 with 7 × 10^6^ spores each). There were three replicates (three Petri dishes per replicate) for each condition.

Guajillo chili seeds were obtained from a commercial source with a guarantee of 92% genetic homogeneity and a germination percentage of 95%. They were surface disinfected in three consecutive steps with three consecutive washes with sterile water after each step: (1) seeds were submerged in 5% NaClO for 1 min with constant agitation, washed with sterile water, and immersed in 70% ethanol for 30 s; (2) seeds were submerged in 2% NaClO for 1 min with constant stirring, washed with sterile water, and immersed in 70% ethanol for 20 s; (3) seeds were submerged in 0.5% NaClO for 1 min with constant stirring, washed with sterile water, and immersed in 70% ethanol for 10 s. After the three steps, seeds were sonicated for 5 min in a 1% Tween 80 (Sigma, St. Louis, MO, USA) solution. Finally, the seeds were dried on sterile absorbent paper [29].

The experimental units were incubated in the dark at 28 ± 1 °C. Daily observations were made for 15 days, during which the number of germinated seeds was recorded (seeds with at least a 2 mm long radicle were considered germinated). From the recorded data, the germination percentage (GP) expressed as the total percentage of germinated seeds at 15 days was determined. The germination speed index (GSI) was calculated according to the formula proposed in [30]:GSI = P1/T1 + P2/T2 + …… + Pn/Tn
where P = number of germinated seeds, T = time in which they germinated, and n = day of the last control. 

The mean germination time (MGT) was calculated according to [31]:MGT = [(x1d1) + (x2d2) + … + (x 15d15)]/x5
where x1, x2, x15 are the seeds germinated on day 1, 2, …, 15; d1, d2, …, d15 are the days of incubation; and x15 is the total number of germinated seeds on day 15 when the final germinated seed count was performed.

### 2.6. Polyphasic Analysis of Selected Strains

The two most efficient organic P-mineralizer and RP-solubilizer strains were further characterized since they also showed the best germination parameters.

Macroscopic and microscopic images with lactophenol cotton blue staining for both fungi were obtained using a Nikon Eclipse Ti-U Microscope with 40× and 60× objectives and were consistent with images related to previously identified strains for *T. virens* and *A. tubingensis*.

For the molecular analysis, spore suspensions from *Trichoderma* sp. BMH-0059 (P mineralizer) and *Aspergillus* sp. BMH-0060 (P solubilizer) were prepared with 1 mL of 0.01 M CaCl_2_·2H_2_O at a density of 1 × 10^6^ spores mL^−1^. Erlenmeyer flasks with 75 mL of PDB liquid medium were inoculated with 1 mL of each spore suspension. The process for genomic DNA extraction was performed as reported in [26]. To characterize these strains more precisely, molecular marker fragments for β-tubulin (Bt2a (5′ GGTAACCAAATCGGTGCTGCTTTC 3′), Bt2b (5’ACCCTCAGTGTAGTGACCCTTGGC 3′)), calmodulin (Cmd5 (5′ CCGAGTACAAGGARGCCTTC 3′), Cmd6 (5′ CCGATRGAGGTCATRACGTGG 3′)), elongation factor 1 (EF1-728F (5′ CATCGAGAAGTTCGAGAAGG 3′), EF1-986R (5′ TACTTGAAGGAACCCTTACC 3′)), and RNA polymerase (RPB2-5F (5′ GAYGAYMGWATCAYTTYGG 3′), RPB2-7CR (5′ CCCATRGCTTGYTTRCCCAT 3′)) were amplified and sequenced. The elongation factor marker could not be amplified for the *Aspergillus* sp. strain.

The PCR protocol for the amplification of the molecular markers was as follows: Fifteen-microliter PCR reactions contained 7.5 μL Q5 High-Fidelity 2X Master Mix (New England Biolabs, Ipswich, MA, USA), 0.75 μL of each 10 μM primer, and 50 ng of genomic DNA. PCR reactions were carried out in a T100 Thermal Cycler (Bio-Rad, Hercules, CA, USA), with the following cycle conditions: initial denaturation at 98 °C for 30 s followed by 35 cycles of 98 °C for 10 s, annealing at 68 °C for 30 s, and extension at 72 °C for 20 s, with a final extension of 72 °C for 2 min, and a hold at 12 °C until recovery of the amplified DNA. The calmodulin and RPB2 markers for *Trichoderma* were amplified with annealing temperatures of 54 and 64 °C, respectively. 

The amplicons obtained from the PCRs for all the markers were cleaned prior to sequencing. For this, 10 μL of PCR product was purified using Exo-CIP Rapid PCR Cleanup Kit (New England Biolabs) according to the manufacturer’s instructions. Then, 3.5 μL of purified PCR product along with 5 μL of 2 μM primer were sent for Sanger sequencing (Eurofins Genomics, Louisville, KY, USA). Sequencing reads were de novo assembled, and the consensus sequence of high-quality base pairs was extracted using Geneious.

For the phylogenetic reconstruction, the sequences were compared and aligned with closely related sequences deposited in the GenBank database using BLAST (https://blast.ncbi.nlm.nih.gov/Blast.cgi; accessed on 1 December 2021), limiting the alignment to only sequences of confirmed type material and excluding uncultured/environmental sample sequences. For strain BMH-0059, concatenated alignments of the DNA sequences for RPB2, elongation factor 1, and ITS were used. For strain BMH-0060, DNA sequences of RPB2, calmodulin, β-tubulin, and ITS (both ITS regions were previously reported in [26]) were used. The sequences were aligned with MAFFT v7.505 in high speed mode (mafft -auto in > out) [32], and the alignments were cured with trimAl tool v.1.4.rev15 under gappyout mode [33]. Phylogenies were estimated with PhyML under the GTR model (option -m 012345) with default approximate Bayes branch supports [34]. Other options were set as default. General clades were grouped to achieve a better final graphic presentation.

### 2.7. Effect of Inoculation of Two Selected Strains on the Growth of C. annuum in Greenhouse Conditions

The effect of the interaction between soil mineralogy and strain inoculation on the growth of *C. annuum* was evaluated. Samples of Andisols and Vertisols were sieved at 4 mm. To observe only the P contribution in these experiments, both kinds of soil were amended with the nutrients that presented deficits (according to Appendix A) in the following way: Andisol was enriched with 0.25 g of (NH_4_)_2_SO_4_, 0.25 g K_2_SO_4_, and 1.5 g Ca (OH)_2_ per kg^−1^ soil and Vertisol with 0.25 g (NH_4_)_2_SO_4_ per kg^−1^ of soil. Then, they were sterilized at 120 °C (1.2 kg cm^−2^ for 20 min for two continuous autoclaving cycles). Five hundred grams of each sterile soil was transferred to 750 mL pots and used for all the experiments. Several treatments were used with both Andisol and Vertisol as follows: An absolute control implied the use of both types of soils without fungal inoculant (WI); an agronomical control was set in which each soil was fertilized with NPK (15:15:15) without fungi to compare the results with a fertilized soil. A third control was set using each soil with acidulated and non-acidulated RP (1:1) to see the effect of soluble P in each soil, but without fungal intervention so as to be able to compare it to the solubilizer strain BMH-0060. The best P-solubilizing fungal strain (*A. tubingensis* BMH-0060) and the best P-mineralizing strain (*T. virens* BMH-0059) were selected for greenhouse experiments.

In all the sterilized substrates, 10 seeds were sown per pot. The treatments with RP contained 2.5 g kg^−1^ of non-acidulated rock phosphate (RP) in 100 g of soil (dry base) and were uniformly mixed with 25 mL with 7 × 10^6^ spores mL^−1^ of the fungal inoculum, according to each treatment. The uninoculated soils received instead 25 mL of the CaCl_2_·2H_2_O 0.01 M sterile solution. In total, 80 experimental units were evaluated (two soils (Andisol and Vertisol), two levels of P (with and without RP), four inoculation treatments (BMH-0059, BMH-0060, BMH-0059 plus BMH-0060, and without inoculum (WI)) and five pots per treatment.

The plants were irrigated daily with deionized water up to 60% field capacity. Every 8 days, the plants were irrigated with a P-free Hoagland solution [35,36], and the foliar P concentration was determined. At four different time points (15, 21, 37, and 45 days post-inoculation), one leaf disc of 0.5 cm Ø from the youngest leaf of each plant per treatment was excised. The leaf discs were incinerated at 500 °C for 24 h. Subsequently, 1 mL of hydrochloric acid (HCl) was added for 20 min to each sample for each treatment. Then, 9 mL of distilled water and 2.5 mL of developer solution were added [37]. The foliar P concentration was determined using the Molybdate Blue method at λ = 890 nm [37] in a spectrophotometer (Genesys 20 ThermoSpectronic, Thermo Fisher Scientific, Waltham, MA USA).

The experiment lasted 45 days, after which all treatments were disassembled. The following growth parameters were evaluated in each experimental unit: (a) number of leaves (No.): count of cotyledonal and true photosynthetically active leaves in the different plants; (b) total length of the plant (cm): the measurement from the root apex to the most apical leaf of each of the sampled plants was considered; (c) stem length (cm): from the longest apical leaf to the root neck; (d) root length (cm): measurement from the apex of the main root to the neck root of each sampled plant; (e) biomass/dry weight (g): plants were dried in an oven at 60 °C for 72 h until reaching constant dry weight. 

### 2.8. Experimental Design and Statistical Analysis

To verify our hypothesis, the experimental design was completely random. For the in vitro germination of *C. annuum* L. seeds, the treatments had a 4 × 4 factorial arrangement (4 strains and 3 doses of inoculum evaluated and 1 uninoculated control). The dependent variables were the seed germination percentage (GP), the germination speed index (GSI), and the mean germination time (MGT). Each treatment had 3 replications.

For the plant growth parameter experiment, the treatments had a 2 × 2 × 5 factorial arrangement (2 soils, 2 levels of P with and without RP, and 4 inoculated treatments and 1 uninoculated control). The dependent variables were biomass and foliar P concentration (mg L^−1^). Each treatment had 4 replicates. The experimental units were distributed randomly.

The results were analyzed through a principal component analysis (PCA), and a two-way ANOVA for which the assumptions in the residuals were determined. All the analyses were carried out with the generic R version 3.4.3. (R Core Team 2017), and the PCA was carried out with the package ade4 version 1.7-15. Duncan’s test was used for the separation of means. The analyses were performed with a level of significance *p* ≤ 0.05, and all analyses were carried out with the statistical software Statgraphics version Centurion XVI. In addition, the correlation between the secretion profile of AOs [26] and the concentration of foliar P was determined.

## 3. Results

### 3.1. Fungal Isolates Present in Different Types of Soils and Root Environments

Since fungal inhabitants from milpas have been scarcely studied, the isolated strains were classified by morphotypification and selective media growth for each collection site. Figure 2 shows representative images of the main genera found in this study. Thirty-one isolates were grouped into 10 morphotypes as judged by macro- and microscopic morphotypification, and these corresponded to *Trichoderma*, *Aspergillus*, *Alternaria*, *Penicillium*, *Rhizopus*, *Mortierella*, *Sclerotium*, *Mucor*, *Paecilomyces*, and *Fusarium*. A very good solubilizing bacterial strain of *Streptomyces* sp. is also shown for comparison (Figure 2). Preliminary P solubilization and mineralization screenings [26] led us to analyze more carefully the CFUs present in the Vertisol of the isolated genera: *Trichoderma*, *Aspergillus*, *Penicillium*, *Rhizopus*, and *Fusarium*, whereas the isolates analyzed from Andisol (that showed less diversity regarding P-solubilizer fungi) were *Trichoderma*, *Penicillium*, and *Aspergillus* (Table 1). Fungal diversity was greatest in Vertisol, and usually more colonies were obtained in the PNM medium, although for certain locations and microhabitats other media had the greatest number of colonies. All the microhabitats showed colonies in the order of 10^4^ CFU per gram of soil (or root, in the case of endophytes). Although a limited number of genera were recovered and could be identified, this is one of the few reports of fungal species inhabiting different “milpas” and different microhabitats.

### 3.2. Biocompatibility among Strains

Since consortia of microorganisms are often more efficient than single strains, the capacity of the 31 isolates to grow in the same dish was evaluated by inoculating a disc with mycelium from each fungal strain to be tested on opposite sides of the same Petri dish. Biocompatibility was defined in those cases where no barriers were observed. From these, seven compatible strains were selected and grouped into five morphologically siilar groups consistent with *Trichoderma* spp. (three strains), *Aspergillus* sp. (one strain), *Penicillium* sp. (one strain), and *Paecilomyces* sp. (one strain). In Figure 1 (in Section 2), representative images of the biocompatibility assays are presented. For this work, it was particularly important to evaluate the compatibility of *Trichoderma* BMH-0059 and *Aspergillus* BMH-0060 and determine if they could grow together (see below).

### 3.3. Selection of P-Solubilizing and -Mineralizing Strains

From the above-mentioned strains, four strains were selected based on preliminary tests for P solubilization and mineralization and their capacity to grow together. BMH-0059 (preliminarily identified as *T. crassum* in [26]), BMH-0060 (preliminarily identified as *Aspergillus awamori* in [26]), BMH-0061 (*Trichoderma* sp.), and BMH-0062 (*Trichoderma pubescens*) were the best performers for P mineralization and solubilization according to a general linear covariance model in two types of medium (OHM and PNM) [26]. These strains were selected for in planta assays.

### 3.4. In Vitro Germination of Chili Seeds

The effect of the concentration and composition of the inoculum on the germination percentage (GP), germination speed index (GSI), and mean germination time (MGT) of chili seeds was measured. A medium concentration of 7 × 10^6^ spores mL^−1^ showed a statistically significant difference (*p* ≤ 0.05) for each strain compared to the control without inoculum and performed better for GP in all cases. GSI showed no statistical difference between medium and high (7 × 10^8^) inocula, except for strains BMH-0061 and BMH-0062, in which the medium concentration performed better. MGT was better with medium inocula except for the treatment with BMH-0060, where high inoculum was statistically similar to medium inoculum (Table 2). However, the best performers were BMH-0059 with 93.1 ± 1.00 GP, 1.192 ± 0.02 GSI, and 5.8 ± 0.15 MGT and BMH-0060 that presented 91.1 ± 1.00 GP, 2.275 ± 0.14 GSI, and 6.9 ± 0.45 MGT (Table 2).

Although strains BMH-0059 and BMH-0060 showed both mineralization and solubilization capacity, BMH-0059 was a better mineralizer and BMH-0060 was a better solubilizer [26], and this correlates roughly with the better performance of these strains observed for GP, GSI, and MGT. In this experiment, dual inoculations were also tested with the best mineralizer and the best solubilizer strains. The co-inoculation of *Trichoderma virens* BMH-0059 and *Aspergillus tubingensis* BMH-0059 (see next section) with medium inoculum showed the greatest synergy for GSI and MGT (*p* ≤ 0.05), while the GP was the same as for both individual strains and in general the best performance for all the treatments.

### 3.5. Polyphasic Identification of the Best Solubilizer and Mineralizer Strains

Given the data obtained in the previous experiments and that *Trichoderma* BMH-0059 and *Aspergillus* BMH-0060 showed dual efficiency in mineralizing phytate and solubilizing RP, further characterization of these strains was performed. Previously, a preliminary molecular characterization of the best four solubilizer strains was performed using only the ITS as a molecular marker, allowing the identification of the strains to the genus level [26]. In this work, the molecular identification of *Trichoderma* sp. BMH-0059 and *Aspergillus* sp. BMH-0060 strains was performed more precisely, as described in Section 2. The *Trichoderma* BMH-0059 (isolated as an endophyte from corn/Alfisol) was consistent with *T. virens* with high branching support and 87% sequence identity (Figure 3. *Aspergillus* BMH-0060 (isolated from the rhizosphere of bean/Andisol) was phenotypically and molecularly consistent with the *A. tubingensis* clade with real branching divisions that accept values with 76% identity for the reference sequences (Figure 3 and Figure 4).

### 3.6. Greenhouse Evaluation of Chili Plant Performance in Different Kinds of Soils

It was confirmed that the mineralogy of the soil and the composition and concentration of the inoculum affects the growth performance of the plants as measured through the amount of foliar P content and dry biomass of chili plants as variables (Figure 5 and Figure 6). 

The general tendency shows that in Andisol soils, the content of foliar P was higher in the condition without RP (Figure 5), and a constant increase until day 45 was recorded. In contrast, in Vertisol soils, the foliar amount of P was higher in the presence of RP and kept increasing as the plants grew. In Andisol, foliar P concentration varied over time with no consistent pattern, independently of the inocula, and was higher or at best equal without RP. The highest concentration of foliar P in this soil was found at day 21 when inoculated with both strains and without RP. In Vertisol with RP, the highest amount of foliar P was achieved at day 15 when the plants were dually inoculated and did not change with time. In contrast, when Vertisol was not added with RP, the best performance regarding foliar P increased with time and was higher when the plants were inoculated with strain BMH-0060 (the best solubilizer) or with dual inoculation as inoculum. The best performance was achieved in all situations with dual strain inoculation of BMH-0060 and BMH-0059 in any of the soils.

The dry biomass of the plants and the growth parameter response (*p* ≤ 0.01) had a close relationship with the concentration of foliar P for all the treatments in both soils (Figure 6, Figure 7, Figure 8 and Figure 9). Again, the best performance was obtained when BMH0059 and BMH0060 combined were used as the inoculum. However, it is worth noting that in Andisol, BMH-0059 (the best mineralizer) performed almost as efficiently as the dual inocula. This shows some correspondence to the soil type P retention characteristics.

A three-way ANOVA was carried out to determine the effect of the factors: “soil”, “RP”, and “inoculum composition” and their interactions on the plant variables: total plant length (cm, TLP), stem length (cm, SL), root length (cm, RL), number of leaves (NL), and total plant dry weight (g, TPDW) (Figure 10). There was no significant difference in the number of leaves (count of cotyledonal and true photosynthetically active leaves in the different plants), total length of the plant (cm), stem length (cm), or root length (cm) among the different treatments.

A principal component analysis (PCA) was carried out to detect the grouping patterns of the plant variables and to reduce the number of variables. The PCA was performed considering TLP, SL, RL, NL, and TPDW as variables.

The PCA analysis was based on the correlation matrix for each principal component. Those plant variables that contributed at least 10% of the variance in the normalized length of the principal component while at the same time having an R ≥ [0.7] were considered significant.

In the case of significant differences in the three-way ANOVA, the Tukey test (*p* < 0.05) was used. Two principal components were responsible for 57.65% of the variance. PC 1 explained 35.21%, while PC 2 explained 22.44%. The variables that correlated significantly with PC1 and PC 2 are shown in Figure 10.

### 3.7. Profile of Organic Acids (OAs) and Its Correlation in Different Soil Types and P Concentrations in Chili Plants

In addition, PCA positively correlated with the OA secretion profile of *T. virens* and *A. tubingensis* in different soils, as reported by Zúñiga Silgado et al. (2020) [26], and foliar P concentration in chili plants grown in Andisol and Vertisol (Figure 11 and Appendix A). According to the results obtained in the solubilization experiments [26], *T. virens* BMH-0059 and *A. tubingensis* BMH-0060 were selected to explore the profile of the secreted OAs in the different soil types and were related to the amount of soluble P in the media. For both types of soils, the concentration of P in solution was similar, with the exception of *T. virens* in Andisol (29.187 ± 0.400 µg mL^−1^) (Appendix A). Although the treatments presented very similar amounts of soluble P in both types of soils, the differences were statistically significant (*p* ≤ 0.01).

Considering the premise that the mineralogy of the soil could be affecting the secretion profile of OAs (types and quantity) and that this would determine the microbial solubilization efficiency of P, a PCA was performed. Based on the coordinates of the main factors that provided the greatest variance, a general linear model was performed that analyzed the effect of the type of soil (Andisol, Vertisol, and without soil (WS)) and the isolates *T. virens* and *A. tubingensis* on the concentration of P in solution and in planta. The coordinates of the mentioned factors generated in the PC analysis were considered as covariates. In Andisol, Vertisol, and treatments without soil, a compact and similar distribution of microorganisms is observed (Figure 10). Based on the GLM model, there is no effect of the type of soil that correlates with the kind and amount of OAs and with the concentration of P in solution.

Figure 11 shows evidence of a differential secretion profile of OAs according to the type of soil (Andisol and Vertisol) and the fungal strains. Most of the tested OAs were secreted in each soil type but in different concentrations, indicating the influence of the soil retention capacity in the type and amount of OAs secreted by each strain. In Andisol, both strains secreted significant amounts of pyruvic, tartaric, and malic acids, while minor concentrations of fumaric, succinic, oxalic, and citric acids were found (Figure 11a, right panel). The only significant difference in this type of soil was that *A. tubingensis* secreted a larger amount of succinic acid than *T. virens*. In Vertisol, the patterns of OA secretion were similar in each strain for pyruvic, oxalic, and citric acids. *A. tubingensis* showed a higher secretion of tartaric > oxalic > pyruvic acids, while, in contrast, no succinic acid was detected for this strain (Figure 11b and Appendix A).

In the control without soil (Figure 11d), only fumaric, succinic, malic, and oxalic acids were detected in very low concentrations when treated with *T. virens* BMH-0059 or *A. tubingensis* BMH-0060, with the exception that *T. virens* BMH-0059 did not secrete succinic acid in this condition. *A. tubingensis* BMH-0060 showed similar levels of fumaric and malic acids to those found with BMH-0059, while succinic acid was the major secreted OA and oxalic acid showed a lower concentration when compared to *T. virens* BMH-0059. The main difference in the cultures without soil is that *T. virens* that BMH-0059 secretes primarily oxalic acid, while *A. tubingensis* BMH-0060 secretes primarily succinic acid, indicating a difference in the type of preferred OA secretion by the different species. The chromatographic profile showed significant differences (*p* ≤ 0.05) for the OAs secreted by *T. virens* BMH-0059 in each of the soils. In this condition, BMH-0059 only secreted fumaric, succinic, malic, and oxalic acids at very low concentrations.

Two main components explained 62.8% of the variance, where PC1 explained 32.28% and PC2 explained 30.51% (Figure 11). The variables that significantly correlated with PC1 were fumaric, tartaric, and citric acids on the negative axis; meanwhile, pyruvic acid and the concentration of P were correlated with the PC2, the first positively and the last negatively (Figure 11a, Appendix A). The two-way ANOVA when considering PC1 as a response variable (which negatively integrates citric, tartaric, and fumaric acids) shows a significant effect of the type of soil (*p* < 0.0001) and the interaction of the soil with the species of fungi (*p* < 0.0001). On the other hand, Appendix A shows the results of the ANOVA considering PC2 as a response variable (which positively integrates pyruvic acid and P concentration on the negative axis) and indicates a significant effect of the type of soil, the species of fungus, and the interaction (soil–fungi) on the phytoabsorption of P in chili.

## 4. Discussion

Previously we demonstrated that the solubilization of RP is favored by the lowering of pH mediated by the secretion of fungal OAs [26]. In the soil, OAs with low acidity constants resulted in fewer negative effects on soil quality, structure, and compaction or moisture retention, among others. In addition, the release of more H^+^ ions due to their low acidity constants (at the same pH as OAs) is a determining factor in their P-solubilizing efficiency [38]. Given that soil mineralogy is usually neglected during in vitro studies, although it determines the efficiency of microbial solubilization of RP [26,39,40], this research confirms that soil mineralogy has a significant influence on the in vitro efficiency of fungal strains in secreting OAs and solubilizing RP. 

In Andisol, contrary to Vertisol, there was a lower concentration of P in solution. Andisol is predominantly composed of amorphous materials (see below) which together with organic matter have a strong P sorption capacity [41]. In Andisol, all the fungi secreted a mixture of OAs to acidify the medium and release P in solution. The results confirm the findings of [26] on how the soil mineralogy and the type of fungus influence the OA secretion profile, and that the efficiency of P solubilization by a microorganism is not directly related to the type or number of OAs released [42].

Regarding the germination of *C. annuum* seeds, *T. virens* and *A. tubingensis* showed a high capacity to improve seed germination. Although several studies have shown that pH is an important factor in the growth and development of fungi and its efficiency in solubilizing inorganic P (Pi) [16,21,43], little is known about the impact of changes in pH mediated by the composition and concentration of fungal inoculum on the germination and growth of chilis. The concentrations of 7 × 10^6^ > 7 × 10^8^ > 7 × 10^4^ spores mL^−1^ positively affected the variables evaluated in all treatments compared with the control (Table 2). Our results agree with those presented in [21,43,44], where a 7 × 10^6^ CFU concentration favored a greater number of total germinated seeds and reduced the average germination time. Higher concentrations (7 × 10^8^ CFU) of inoculum are likely to increase competition for limited resources, while lower concentrations (7 × 10^4^ CFU) of inoculum do not reach the load capacity required to activate metabolic and genetic signals important to initiate germination. Both *Trichoderma* spp. and *Aspergillus* spp. in their role as phosphorus-solubilizing fungi (PSFs) and phosphorus-mineralizing fungi (PMFs) are known to stimulate both nutrition and plant health [8,17,20]. In synergy, these fungi were probably capable of significantly increasing the degradation of the seed coat through the action of lignocellulolytic enzymes, producing compounds such as monomeric hexoses (glucose, mannose, and galactose) and pentoses (xylose and arabinose) that are an important carbon source for the seed, as previously reported [45]. In addition, fungi produce various secondary metabolites such as auxins, cytokinins, and gibberellins [46] which when released into the environment can stimulate germination and plant development [17,21].

Although it is known that soil mineralogy affects the ecophysiological response of plants, for the first time this study shows the effect of the tripartite interaction of soil type, RP, and inoculum composition on the response of P absorption, biometry, and accumulation of dry mass in chili plants. In Andisols, the compaction of the soil increases its mechanical resistance, reducing the volume that can be explored by the roots and thus limiting spatial access to nutrients and water. This process is more important for nutrients such as P that have little mobility in the soil. The transport of nutrients from the soil to the roots is carried out by two main mechanisms: mass flow and diffusion. However, in the case of poorly mobile nutrients such as P, only small amounts reach the roots by mass flow, although some diffusion could also be possible. So, since a robust root system was not developed in this type of soil, the growth parameter variables evaluated in chili in this type of soil were lower than those in Vertisol.

In Andisol, we expected a decrease in P availability as a function of time due to the negative effect of soil compaction, oxygen diffusion rate, and water mobility [41,47]. Based on this premise, we expected that the availability on day 15 would be greater than that on day 45. As expected, this trend was observed and contrasted with the response of the plants in all treatments in Vertisol. In Andisol, contrary to Vertisol, the addition of RP did not improve the porous system or its apparent density (APD), but rather compacted it, preventing adequate radical development. Thus, a better APD in Vertisol would have a greater availability of phosphate ions in solution and therefore a greater amount of absorbed P [48]. Our data were consistent with another report [19], showing a higher concentration of foliar P, biometry, and dry biomass of chili plants in all treatments including controls. Given that other groups [16,49,50] demonstrated that the type of soil can be related to the activity of microorganisms, we evaluated the adequate selection of co-inoculants according to the mineralogy of Andisol and Vertisol. In Andisol, amorphous materials such as allophane, imogolite, and ferrihydrite clays and complex mineral structures of iron phosphates (Fe^3+^), aluminum (Al^3+^), or organic matter (OM) predominate and strongly absorb P [40]. Under these conditions, fungi that solubilize Pi secrete greater amounts of OAs, strongly acidifying the rhizosphere, while fungi that mineralize organic P (Po) secrete enzymes (phosphatases) to release Po [51,52]. In this sense, both (Al^3+^) and (Fe^3+^) are inhibitors of various enzymes of the Krebs cycle and of phytases [21,53], which would explain our results.

In our analysis, two components explained 62.8% of the variance. PC1 explained 32.28%, while PC2 explained 30.51%. The variables that significantly correlated with PC1 were fumaric, tartaric, citric, and oxalic acids on the negative axis, meanwhile pyruvic and malic acids were positively correlated with PC2, and P concentration was negatively correlated. The two-way ANOVA when considering PC1 as a response variable (which negatively integrates citric, tartaric, oxalic, and fumaric acids) shows a significant effect of the type of soil (*p* < 0.0001) and the interaction of the soil with the species of fungi (*p* < 0.0001). 

Another additional aspect to consider is that depending on their quantity and reactivity, exchangeable bases such as Ca^2+^ and Mg^2+^ can be cytotoxic to both the fungus and the plant by blocking enzymatic pathways important for the secretion of OAs and phytases, thereby negatively affecting the bioavailability and phytoabsorption of P [20,43]. *T. virens* BMH-0059 was positive for the amplification of a portion of the *phyA* gene, confirming its mineralizing performance (manuscript in prepapration). On the other hand, in Andisols, it is possible that the secreted OAs have preferentially formed complexes with the exchangeable Ca^2+^ rather than with the Ca^2+^ of RP. In Vertisol, the released Ca^2 +^ ions were probably precipitated or chelated by carboxylic acids/anions (oxalate or citrate). If so, this would have generated a dynamic equilibrium that favored the continued dissolution of the RP. Under these conditions, Ca^2+^ (precipitated or chelated) did not produce allosteric inhibition of the enzymes involved in the production of OAs or phosphatases; therefore, a higher concentration of P in the solution was found [54,55]. Our findings are consistent with those of Bononi et al. (2020) [16], who studied the solubilization of RP in the *Trichoderma*–soybean interaction; with Pelagio-Flores et al. (2020) [21], who evaluated the acidification of the rhizosphere by secretion of OAs in the interaction of *Trichoderma atroviride*–*Arabidopsis thaliana*; and with Redel et al. (2008) [51], who studied the extracellular phytases in the *Aspergillus niger*–*Arabidopsis thaliana* interaction. All these investigations agree that the increase in rhizospheric acidity negatively affects the growth of roots and ultimately hinders the seed germination of different plants. According to our results, the inhibition of the germination and growth of chili seedlings in Andisol when inoculated with *T. virens* and/or *A. tubingensis* could be explained by an increase in rhizospheric acidity combined with the physical–chemical properties inherent to the soil type. This significantly diminishes the growth variables evaluated [56]. Thus, according to Bononi et al. (2020) [16], the low concentration of P in the soil reflects a decrease in the production of ATP and NADPH and the expression of genes related to photosynthesis and synthesis of phytases, strongly affecting the growth variables evaluated.

Interestingly, in Vertisol, the interaction of chili with *T. virens* and/or *A. tubingensis* caused a significant increase in root elongation, P capture, germination, and plant growth promotion. In this regard, recent findings reported by Pelagio-Flores et al. (2020) [21] suggest that the growth promotion induced by *Trichoderma* spp. during the early stages of the plant–fungus interaction could be attributed mainly to volatile organic compounds (VOCs) rather than the release of auxins or other diffusible compounds by the fungus. Vertisols present an adequate APD that affects better VOC diffusion. In other reports, [57,58] the exposure of the plants to the VOC of *T. virens* could have stimulated plant growth, chlorophyll content, and plant biomass and size, which could be correlated with a greater capacity of exploration of the soil, better rooting, and a greater capacity to absorb nutrients and water.

## 5. Conclusions

For the first time, a systematic isolation of fungi from milpa soils was conducted. This contributed to the knowledge of fungal genera present in this pre-Hispanic agroecological system. We could select fungal strains capable of mobilizing insoluble phosphorus to obtain a better landscape of how the soil type affects the nourishment of P in chili plants. In particular, a *T. virens* strain and an *A. tubingensis* strain showed both mineralizing and solubilizing activities; however, *T. virens* was more efficient in mineralizing P, and *A. tubingensis* was more efficient in solubilizing it. This resulted in different effects on chili plant development when adding RP to two different kinds of soils with different retention capacities. To our knowledge, this is the first report of the tripartite interaction among soil type, composition, and concentration of inoculum and the presence/absence of RP. Depending on the soil type, the addition of specific fungi could potentially enhance P bioavailability, promoting germination, growth, and development of agriculturally important crops. In view of this, it should be specified that the diagnosis of physical and chemical fertility should be considered before defining the availability of nutrients, and eventually, corrective practices such as biological fertilization could be recommended [19].

## Figures and Tables

**Figure 1 jof-08-01178-f001:**
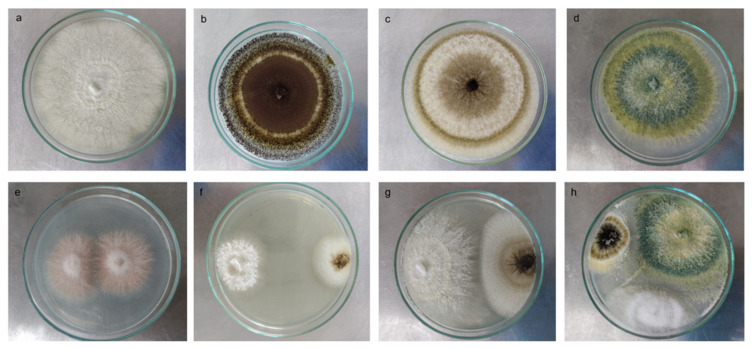
Biocompatibility test between P-solubilizing and P-mineralizing fungal strains. Macroscopic individual images of the colony: (**a**) *Paecillomyces* sp., (**b**,**c**) *Aspergillus* spp., (**d**) *Trichoderma* sp. Macroscopic example images of biocompatibility between (**e**) *Fusarium* strains, (**f**,**g**) *Aspergillus* sp., and *Trichoderma* sp. (**h**) Triple interaction between *Aspergillus* sp., *Trichoderma* sp., and *Paecillomyces* sp. The square shows a 60× approximation of the mycelium in the culture medium.

**Figure 2 jof-08-01178-f002:**
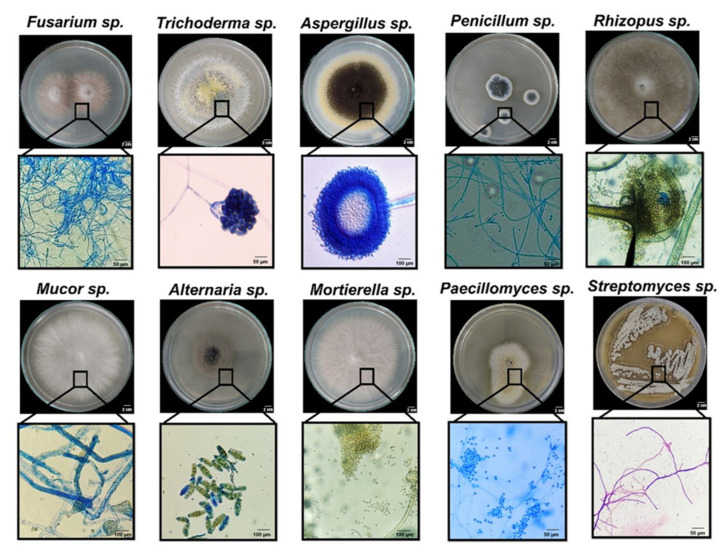
Representative images of the fungal genera encountered in two different milpas with different soil: Vertisol and Andisol. Lower panels below each of the genera show magnification of conidia and hyphae (see Table 1 for the origin of the strains).

**Figure 3 jof-08-01178-f003:**
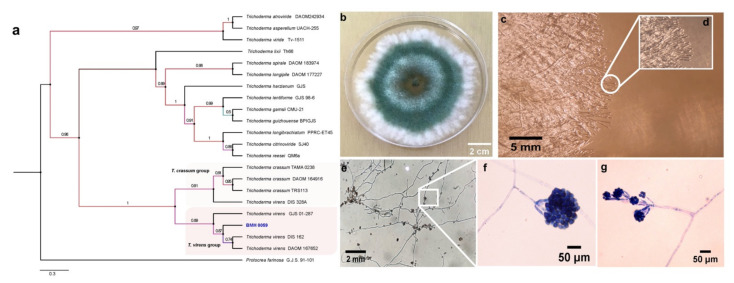
Phylogenetic tree generated by FastTree analysis using a MAFFT alignment of ITS, β-tubulin, calmodulin, EF1, and RPB2 nucleotide sequences obtained from type material. The tree includes strains related to (**a**) strain BMH-0059 (**left**) identified as *T. virens*. Bootstrap values (>50%) are labeled in color on the branch nodes. Representative images of the colony are shown on the right side of each alignment. (**b**) Macroscopic image of the colony. (**c**–**g**) Hyphal growth of the selected strain. (**c**–**e**) The square shows a 60× amplification of the mycelium in the culture medium. (**e**–**g**) Reproductive structures of the strain at different augmentations (conidiophores).

**Figure 4 jof-08-01178-f004:**
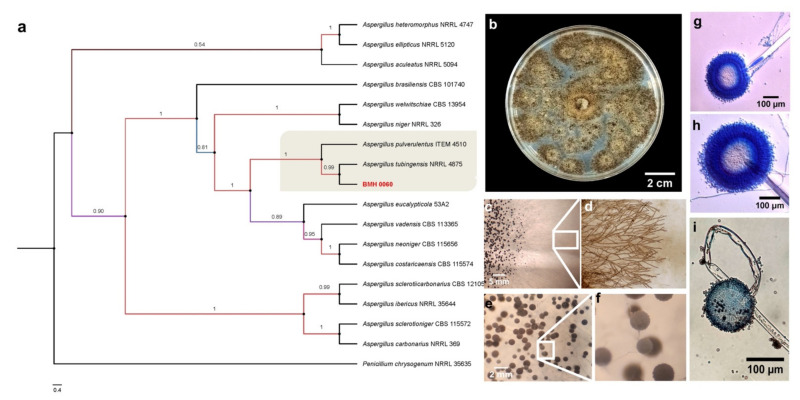
Phylogenetic tree generated by FastTree analysis using a MAFFT alignment of ITS, β-tubulin, calmodulin, EF1, and RPB2 nucleotide sequences obtained from type material. The tree includes strains related to (**a**) strain BMH-0060 (**left**) identified as *A. tubingensis*. Bootstrap values (>50%) are labeled in color on the branch nodes. Representative images of the colony are shown on the right side of each alignment. (**b**) Macroscopic image of the colony. (**c**–**i**) Hyphal growth of the selected strain. (**c**–**e**) The square shows a 60× amplification of the mycelium in the culture medium. (**e**–**g**) Reproductive structures of the strain at different augmentations (conidiophores).

**Figure 5 jof-08-01178-f005:**
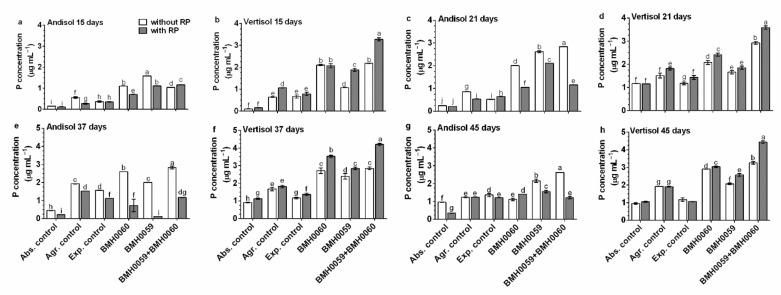
Concentration of foliar P in plants with different treatments. *C. annuum* plants were evaluated with chemical treatments (without RP, white bars; with RP, black bars). Single and dual inoculations were performed with *T. virens* (P-mineralizing fungus (PMF)) and *A. tubingensis* (P-solubilizing fungus (PSF)). In addition, three different controls were used: without inoculum (Abs. control) (without inoculum, without RP or NPK), agronomical control (Agr. control) (without inoculum, with NPK), and experimental control (Exp. control) (without inoculum, with NPK). The inoculum contained 7 × 10^6^ spores mL^−1^. Concentration of leaf P was evaluated at 15, 21, 37, and 45 days and analyzed by ANOVA multiple comparison tests. Bars represent the standard error. Letters above the bars indicate significant differences. All values shown are the average of 5 replicates per treatment.

**Figure 6 jof-08-01178-f006:**
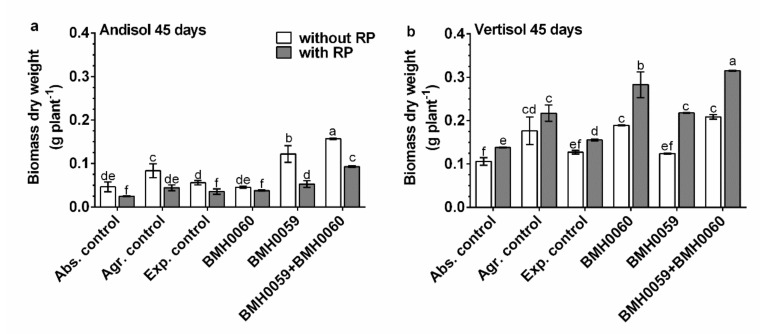
*Capsicum annuum* L. with chemical treatments (NPK and RP), inoculated singly and dually with *T. virens* (PMF) and *A. tubingensis* (PSF) (mineralizers and solubilizers of P fungal at a rate of 7 × 10^6^ spores mL^−1^). In addition, three different controls were used: without inoculum (Abs. control) (without inoculum, without RP or NPK), agronomical control (Agr. control) (without inoculum, with NPK), and experimental control (Exp. control) (without inoculum, with NPK. Dry biomass of the plants was evaluated at 45 days and analyzed by ANOVA multiple comparison tests. Bars represent the standard error. Letters above the bars indicate significant differences. All values shown are the average of four replicates per treatment.

**Figure 7 jof-08-01178-f007:**
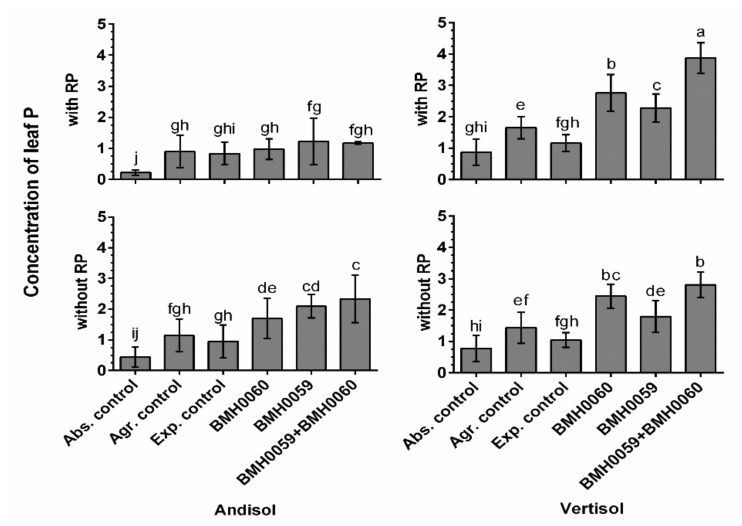
Concentration of leaf P of *C. annuum* L. plants grown in soil with and without RP and inoculated with *T. virens* (PMF) and/or *A. tubingensis* (PSF) was measured. “Soil”, “RP”, and “inoculum composition” were treated as factors in the ANOVA. Covariance of the time in which the P measurements were recorded was considered. Significant differences between factors shown by lowercase letters were determined by Tukey test (*p* < 0.01).

**Figure 8 jof-08-01178-f008:**
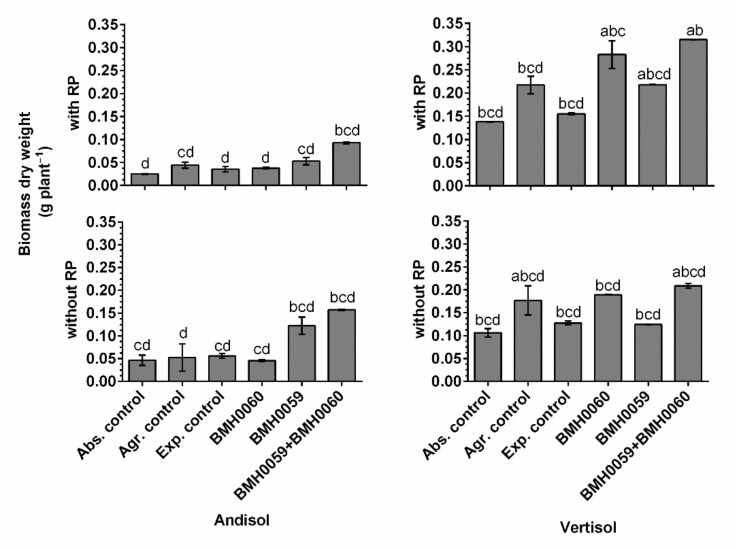
Biomass dry weight (grams per plant) of *C. annuum* L. plants grown in soil with and without RP and inoculated with *T. virens* (PMF) and/or *A. tubingensis* (PSF) was measured. “Soil”, “RP”, and “inoculum composition” were treated as factors in the ANOVA. Covariance of the time in which the total dry plant biomass was recorded was considered. Significant differences between factors shown by lowercase letters were determined by Tukey test (*p* < 0.01).

**Figure 9 jof-08-01178-f009:**
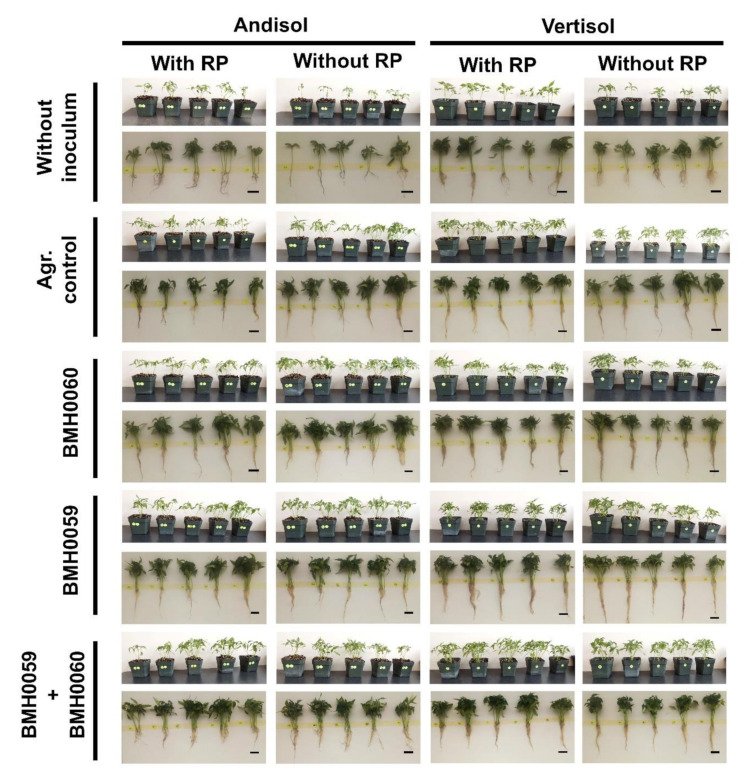
Representative images of the growth parameters in *C. annuum* L. inoculated with and without RP (RP^+^ and RP^−^), adding single and dual treatments with *T. virens* (PMF) and/or *A. tubingensis* (PSF) (7 × 10^6^ spores mL^−1^), grown in two soils (Andisol (And) and Vertisol (Ver)).

**Figure 10 jof-08-01178-f010:**
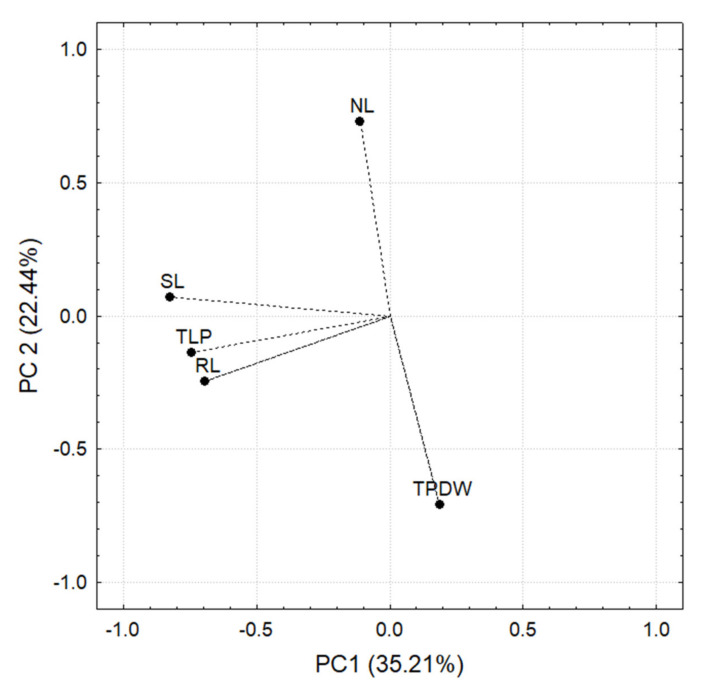
Principal components based on the correlation matrix of the measurements carried out in *C. annuum* L. plants. Note that SL, TLP, and RL negatively correlate with PC 1, while with PC 2 TPDW is negatively correlated and NL is positively correlated.

**Figure 11 jof-08-01178-f011:**
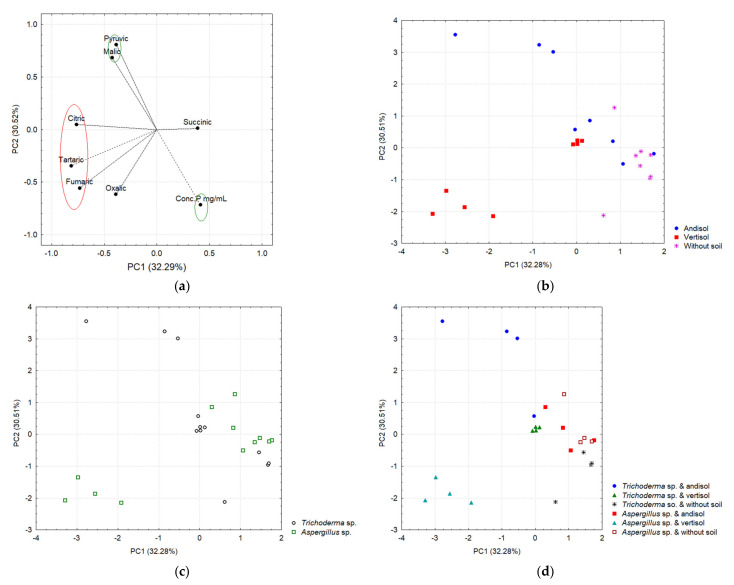
Main components based on the acid correlation. (**a**) In the red ellipse are the organic acids that correlated with PC1, while the green ellipses indicate those variables that correlated with PC2. (**b**) Separation of acids as a function of the soil. (**c**) Separation of acids as a function of fungi. (**d**) Separation of acids as a function of interaction.

**Table 1 jof-08-01178-t001:** Composition of the rhizosphere, rhizoplane, and endophytic fungal community of Vertisol and Andisol.

Substratum	Culture Media	Microbial Genera Present in Vertisol (CFU g^−1^)							Fungal Genera Present in Andisol (CFU g^−1^)		
		*Trichoderma*	*Aspergillus*	*Penicillium*	*Paecilomyces*	*Streptomyces*	*Rhizopus*	*Fusarium*	*Trichoderma*	*Aspergillus*	*Penicillium*
Rhizosphere	PDA	6.5 × 10^4^	5.4 × 10^4^	3.9 × 10^4^	3.3 × 10^4^	2.4 x10^4^	2.5 × 10^4^	1.4 × 10^4^	2.5 × 10^4^	1.4 × 10^4^	1.9 × 10^4^
	BRA	7.8 × 10^4^	6.0 × 10^4^	3.4 × 10^4^	3.5 × 10^4^	2.0 × 10^4^	-	1.5 × 10^4^	2.5 × 10^4^	1.5 × 10^4^	1.4 × 10^4^
	OHM	4.5 × 10^4^	4.5 × 10^4^	5.0 × 10^4^	4.0 × 10^3^	2.5 × 10^4^	6.0 × 10^4^	1.5 × 10^4^	6.0 × 10^4^	1.5 × 10^4^	1.5 × 10^4^
	PNM	8.5 × 10^4^	7.3 × 10^4^	7.5 × 10^4^	4.2 × 10^4^	2.3 × 10^4^	5.5 × 10^4^	1.3 × 10^4^	5.5 × 10^4^	1.3 × 10^4^	1.5 × 10^4^
Rhizoplane	PDA	1.8 × 10^4^	2.5 × 10^4^	2.5 × 10^4^	1.7 × 10^4^	1.5 × 10^4^	2.8 × 10^4^	4.5 × 10^4^	2.8 × 10^4^	4.5 × 10^4^	4.5 × 10^4^
	BRA	1.0 × 10^4^	2.0 × 10^4^	2.9 × 10^4^	1.3 × 10^4^	1.0 × 10^4^	-	-	5.5 × 10^4^	5.0 × 10^4^	4.9 × 10^4^
	OHM	1.5 × 10^4^	2.5 × 10^4^	1.2 × 10^4^	1.0 × 10^3^	1.5 × 10^4^	5.0 × 10^4^	4.5 × 10^4^	5.0 × 10^4^	4.5 × 10^4^	4.0 × 10^4^
	PNM	1.8 × 10^4^	1.5 × 10^4^	1.5 × 10^4^	1.7 × 10^3^	1.5 × 10^4^	5.0 × 10^4^	5.0 × 10^4^	5.0 × 10^4^	1.0 × 10^4^	4.0 × 10^4^
Endophyte	PDA	-	-	-	-	-	-	1.4 × 10^4^	-	-	-
	BRA	1.8 × 10^4^	-	-	-	-	-	-	1.5 × 10^4^	-	-
	OHM	1.5 × 10^4^	-	1.0 × 10^4^	-	1.5 × 10^4^	-	1.3 × 10^4^	1.0 × 10^4^	-	-
	PNM	1.5 × 10^4^	-	-	-	1.3 × 10^4^	-	1.0 × 10^4^	1.0 × 10^4^	-	-

Shaded boxes indicate the greatest number of colonies found for that species in that soil type.

**Table 2 jof-08-01178-t002:** Evaluation of the effect of the composition and concentration of the inoculum of *T*. *virens* and *A*. *tubingensis* on the germination percentage (GP) (%), germination speed index (GSI), and mean germination time (MGT) of chili seeds. ANOVA multiple comparison tests: (a) four inoculation treatments and (b) three inoculum concentrations, 7 × 10^4^ (low), 7 × 10^6^ (medium) and 7 × 10^8^ (high) spores mL^−1^, of each fungal strain with P-solubilizing and -mineralizing capacity. All values are the mean ± SD values of four replicates per treatment (*p* ≤ 0.05).

Strain(s)	Treatment	GP	GSI	MGT
Without Inoculum	Control	53.4 ± 1.00 ^d^	0.994 ± 0.003 ^e^	9.13 ± 0.03 ^a^
BMH-0059 *T. virens*	Low inoculum	44.1 ± 0.98 ^e^	1.325 ± 0.10 ^d^	7.20 ± 0.10 ^b^
	Medium inoculum	93.1 ± 1.00 ^a^	1.962 ± 0.02 ^c^	5.80 ± 0.15 ^e^
	High inoculum	64.4 ± 1.00 ^c^	1.725 ± 0.10 ^c^	6.70 ± 0.10 ^c^
BMH-0061 *Trichoderma* sp.	Low inoculum	53.0 ± 1.00 ^d^	0.990 ± 0.11 ^e^	9.09 ± 0.06 ^a^
	Medium inoculum	93.3 ± 1.00 ^a^	1.849 ± 0.10 ^c^	6.17 ± 0.09 ^d^
	High inoculum	52.3 ± 1.17 ^d^	0.999 ± 0.03 ^e^	9.00 ± 0.00 ^a^
BMH-0062 *T. pubescens*	Low inoculum	60.7 ± 1.00 ^c^	1.221 ± 0.05 ^d^	7.40 ± 0.10 ^b^
	Medium inoculum	91.7 ± 0,70 ^a^	2.621 ± 0.11 ^a^	5.90 ± 0.15 ^e^
	High inoculum	66.7 ± 1.00 ^b^	1.821 ± 0.10 ^c^	6.60 ± 0.10 ^c^
BMH-0060 *A. tubingensis*	Low inoculum	53.8 ± 0.64 ^d^	1.294 ± 0.10 ^d^	9.14 ± 0.03 ^a^
	Medium inoculum ^6^	91.1 ± 1.00 ^a^	2.275 ± 0.14 ^b^	6.90 ± 0.45 ^c^
	High inoculum	65.7 ± 1.00 ^bc^	1.828 ± 0.10 ^c^	6.50 ± 0.00 ^c^
BMH-0059 + BMH-0060	Medium inoculum each	93.1 ± 1.00 ^a^	1.662 ± 0.15 ^c^	4.80 ± 0.10 ^f^

All values shown are the average of four replicates per treatment (*p* ≤ 0.05). Shaded boxes indicate the best performers for each parameter. The super index letters indicate statistically significant differences.

## Data Availability

All the data supporting the findings of this study are included in this article.

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
