# Peer review of "Soil Type Influences Novel “Milpa” Isolates of Trichoderma virens and Aspergillus tubingensis That Promote Solubilization, Mineralization, and Phytoabsorption of Phosphorus in Capsicum annuum L."

_jof, 2022, doi:10.3390/jof8111178_

Round 1

Reviewer 1 Report

The current manuscript "“Milpa” isolates of Trichoderma virens and Aspergillus tubingensis promote solubilization, mineralization and phytoabsorption of phosphorus in Capsicum annuum L depending on soil type" I think this manuscript can only be published after justify some points:

-        The abstract part needs to rewrite in a way to define the exact novelty and originality of your work.

-         Lines 33-34 "In addition, there are several studies regarding its bactericidal, anticancer, and antidiabetic activities."?

-         All abbreviations used should be mentioned in the place of their first mention followed by an abbreviation and then only the abbreviation is written.

-        The introduction must be completed by clarifying the main objectives of the research and by motivating the experimental strategy adopted by authors.

-          Please specify the novelty of your study in the introduction.

-      There are many sentences in the introduction without references. Authors should update references.

-          Authors should move Figure 1 into the results.

-          Add references to all methods.

-          Replace Figures 3, 4 and 10 with clearer images.

-      The conclusion is poorly; I think author should try to link better their work; I mean, the results should be quantitatively reported to present these potential applications better.

-       The whole manuscript must be checked to avoid the presentation of same information several times.

-        The English language needs to be significantly improved, in wording, grammar and sentence structure.

-          I suggest the authors to go through the manuscript one more time to minimize some errors, typos etc.

Author Response

Reviewer 1:

The current manuscript "“Milpa” isolates of Trichoderma virens and Aspergillus tubingensis promote solubilization, mineralization and phytoabsorption of phosphorus in Capsicum annuum L depending on soil type" I think this manuscript can only be published after justify some points:

-        The abstract part needs to rewrite in a way to define the exact novelty and originality of your work.

The abstract has been modified accordingly

-         Lines 33-34 "In addition, there are several studies regarding its bactericidal, anticancer, and antidiabetic activities."?

Yes, chili has these properties, here are some examples:

Chilczuk, Barbara, Beata Marciniak, Anna Stochmal, Łukasz Pecio, Renata Kontek, Izabella Jackowska, and Małgorzata Materska. 2020. "Anticancer Potential and Capsianosides Identification in Lipophilic Fraction of Sweet Pepper (Capsicum annuum L.)" Molecules 25, no. 13: 3097. https://doi.org/10.3390/molecules25133097

Mohammed, A., Koorbanally, N., & Md, S. I. (2017). Anti-diabetic effect of Capsicum annuum L. fruit acetone fraction in a type 2 diabetes model of rats. Acta poloniae pharmaceutica74(6), 1767-79.

Ekom, S. E., Tamokou, J. D. D., & Kuete, V. (2021). Antibacterial and Therapeutic Potentials of the Capsicum annuum Extract against Infected Wound in a Rat Model with Its Mechanisms of Antibacterial Action. BioMed research international2021.

We did not include the references in the  Introduction text because the aim of or work does not deal with these properties. Our work is focused on phosphorous (P) uptake in chili plants with the help of two fungal isolates. However, in the abstract the mention of these properties stresses the importance of achieving good growth conditions of these plants in different kinds of soils, that have different retention potential of P.

-         All abbreviations used should be mentioned in the place of their first mention followed by an abbreviation and then only the abbreviation is written.

This has been corrected

-        The introduction must be completed by clarifying the main objectives of the research and by motivating the experimental strategy adopted by authors.

The Introduction has been modified accordingly

-          Please specify the novelty of your study in the introduction.

In lines 105-106 of the revised version, we specify that there is little knowledge on how the mineralogy of the soil can affect the plant microbe interactions. In the las paragraph, we clearly state the goal of this study

-      There are many sentences in the introduction without references. Authors should update references.

Now the most relevant references are in the Introduction

-          Authors should move Figure 1 into the results.

We are sorry to disagree with this suggestion. The other two reviewers agreed with this Figure being in the Materials and Methods section. It is important that in the Materials and Methods section the representative images of the method are shown. A lot of this confrontations are not shown, since the goal of this experiments was to get isolate combinations that could act in consortium. The relevant combination regarding biocompatibility and P solubilizing and mineralizing capacities was tested. The last two properties were reported in a previous work, recently published:

Zúñiga-Silgado, D.; Rivera-Leyva, J.C.; Coleman, J.J.; Sánchez-Reyes, A.; Valencia-Díaz, S.; Serrano, M.; de-Bashan, L.E.; Folch-Mallol, J.L. Soil Type Affects Organic Acid Production and Phosphorus Solubilization Efficiency Mediated by Several Native Fungal Strains from Mexico. Microorganisms. 2020, 8, 1337; doi:10.3390/microorganisms8091337.

-          Add references to all methods.

Most of the methods described in this work were developed by our group, since, precisely, there are very few studies that consider soil mineralogy in the fungal-plant interactions. When pertinent, we did include the references.

-          Replace Figures 3, 4 and 10 with clearer images.

The figures have been enhanced in resolution; we hope they are now satisfactory

-      The conclusion is poorly; I think author should try to link better their work; I mean, the results should be quantitatively reported to present these potential applications better.

The conclusion has been modified accordingly

-       The whole manuscript must be checked to avoid the presentation of same information several times.

This has been revised, in many sections text has been deleted.

-        The English language needs to be significantly improved, in wording, grammar and sentence structure.

Two of the co-authors are native English speakers, they have revised the text for these issues.

-          I suggest the authors to go through the manuscript one more time to minimize some errors, typos etc.

Done

Thank you for your suggestions

Submission Date

07 October 2022

Date of this review

22 Oct 2022 16:01:17

Answers to Reviewer 2:

We want to thank you for your comments, they will undoubtedly make our work better

Topic is divided in discontinues sections.

This is the format required by the Journal (1. Introduction, 2: Materials and Methods, which can be subdivided in specific methods (2.1; 2.2; 2.3, etc.); Results, which can also be divided in different groups of experiments according to the topic, etc.

Revise the title to make the meaning clear 

The title has been modified

Abstract should contain some numerical data. Abstract should be started with a concise introduction or background of your title or objectives not a general introduction of only plant

The abstract has been modified as requested by the reviewer

Words already present in title should be avoided to add in keywords

The Key words have been modified

Second paragraph should be merged with first one

Done

I did not find the repetition in green house experiment

This was presented as supplementary material, now it is a Figure is included in the main text

 3.6. Greenhouse evaluation of chili plants..... meaning is not clear.... did you evaluate chili plants  ?

We have changed the subtitle, now it reads “3.6. Greenhouse evaluation of chili plant performance in different kinds of soils”

No test was applied to data in Figure 5 and 6 to know the significant difference among different treatments

Figures 7 and 8 show the statistical tests of the data presented in Figures 5 and 6. (so the figures would not be too “crowded”). Nevertheless, we have modified figures 5 and 6 which now show above the bars the ANOVA significance values

Conclusion is too short. Conclude the whole findings in this section 

The Conclusion section has been modified accordingly.

Reviewer 3:

Comments and Suggestions for Authors

Comments to the author JoF (1986478)

The manuscript entitled ““Milpa” isolates of Trichoderma virens and Aspergillus tubingensis promote solubilization, mineralization and phyto absorption of phosphorus in Capsicum annuum L depending on soil type” is really (really.!) interesting one and the author (s) performed a nice work. I truly enjoyed the reading of the manuscript. The study by Zúñiga-Silgado et al. demonstrated the use of Trichoderma and Aspergillus as promoter for capsicum. Overall, the manuscript is well structured; presenting novelty and authenticity of work. The results are reliable and manuscript is in accordance with the Journal’s scope. However for the improvements, major revision is compulsory before further consideration.

In abstract, the author mentioned the methodology; it should not be the part of abstract, for example:

Line 37-40,

We have deleted this sentence

There are no results in abstract only the line 41-43

The whole abstract has been modified, so now it provides a better context of our work and describes the main findings and concludes by suggesting this approach would be useful to decide fertilization on different soil types

Further, in abstract, the conclusion of work must be concise and possible future aspects should be illustrated.

Introduction

See above (…concludes by suggesting this approach would be useful to decide fertilization on different soil types)

Line 49: and à which belongs

Corrected

Line 55-56 à dollar $

Corrected

Line 56-57 repeating sentence in abstract and intro

The sentence was deleted from the abstract.

Line 59-60: “thus is one of the most critical factors limiting plant growth” what does it means which factor?

“factor” has been replaced by “element”

Line 76-77: repeating sentences with line 63-64

The sentence in lines 63-64 has been deleted

Line 77: agronomic management or Agronomic practices?

We changed “management” for “practices “as suggested

Line 79: satisfy à use appropriate word

Has been changed for “suffice”

Line 81-85: Too long sentence

The sentence has been split in two and some non-relevant information has been deleted. Now it reads:

“Faced with this situation, the need arises to implement low-cost technological alternatives to improve the acquisition of P by plants. The use of rhizospheric microorganisms with the capacity to solubilize and mineralize P has proven to be an alternative [14, 15]. from chemically unavailable forms such as RP and organic material respectively

Line 86: changes in the soil (what kind of changes?)

“such as pH, water retention, ionic force, etc..” has been added to the text.

Line 92-93: “Microbial consortia have been used successfully as biofertilizers in various crops of agronomic importance in Mexico such is the case of chili” à Microbial consortia as biofertilizers have been successfully used in various crops of agronomic importance including chili in Mexico.

The sentence has been modified as suggested

Line 98-100: Unclear sentence, need to rephrase

This whole section has been rephrased

Line 104-107: long sentence, need appropriate rephrasing

The sentence has been rephrased; we hope that now is clearer

General comments for introduction:  

Insufficient information for the utilization of Trichoderma and Aspergillus as biofertilizer and/or as plant growth promoter.

We added the following sentence in (now) lines 93-102:

The Trichoderma genus is one of the most studied fungi for its plant-growth promoting activity [16, 17]. Its plant-growth promoting effect is due to several properties such as being a mycoparasite which keeps control on fungal diseases in plants, it triggers the defence mechanisms of the plant via jasmonic and salicylic acids, as well as the ethylene pathway. Finally, it has been shown that they acidify the soil making insoluble nutrients available to the plant and modifying root architecture. Aspergillus spp. have not been studied as profoundly as Trichoderma, but there are a few reports in which it has been shown to secrete organic acids, thus making nutrients more available to the plant. [18]. These fungi such as Trichoderma [16, 17] and Aspergillus [18] are capable of solubilizing…

Materials and methods

Section 2.2. No information regarding the incubation of strains, either fungal or bacterial. As the author mentioned PDA, I believe that for the isolation different fungal and bacterial microorganisms are present as these are 300 colonies. Thus, need to clearly elaborate this section. 

We have added the required information in lines 149-155 of the revised version

All the media were supplemented with ampicillin (100 mg mL-1) and Cloramphenicol (30 mg mL-1) to avoid bacterial growth. The dishes were incubated at 28 °C until fungal growth was observed. The first selection was carried out in the mineralizing medium (PNM) and the solubilization medium (OHM). From the colonies that grew in these media, three monosporic cultures were performed in PDA for each strain to ensure purity of the isolate. BRA medium was used for selection of Trichoderma spp. and to differentiate it from Rhizopus growth. In all cases in the initial … 10-6 dilution approximately 300 colonies were observed, and these numbers were used to make the calculations of CFU per gram of soil or root tissue in the case of endophytes

Section 2.4: the author tested thirty one isolates, but later on only four strains were mentioned (line 149). There is a confusion that how these strains were selected and either these strains were among the thirty one?

We changed the sentence to make clear how these strains were selected:

“Based on their P mineralization and solubilization capacity, as well as their biocompatibility, from the 31 isolates only strains BMH-0059, BMH-0060, BMH-0061, and BMH-0062 were further studied”

Section 2.5: How the seeds were inoculated, by spraying or seed priming method? And he most important, how long (time duration) the seed inoculation was?

This has been more clearly described in the materials and methods section:

“Seeds were imbibed in a spore 7 x 106 solution for each fungus, were shaken for 8 h. at 100rpm. then the seeds were transferred to wet paper towels with Hoagland medium until germination.”

How the spore suspension was prepared and how the spores’ suspension was adjusted to working concentrations?

The spores were collected from the Petri dishes as stated and using a Newbauer chamber volumes were adjusted to get the final concentrations used in the experiment. So now, line 163 (in the revised version) reads:

“To evaluate the effect of the inoculum concentration on the germination of chili seeds, spore suspensions were prepared adjusting the volumes using a Newbauer chamber from the freshly grown strains in PDA collected in CaCl2 • 2H2O 0.01M at three different concentrations: 7×104, 7×106, and 7×108 spores mL-1 of each fungal strain.”

As well, for clarity, lines 165-167 now read:

“Ten chili seeds per Petri dish were inoculated with 5 mL of each spore suspension. In each Petri dish a cellulose membrane was placed.”

The three replicas are mentioned in line 170 of the revised version:

“Three replicas (three Petri dishes/replica) for each condition were performed.”

What is experimental unit? Containing? Petri plates/pots ?

For the germination experiments Petri Dishes were the experimental unit as stated in line 170 of the revised version. The experimental units for plants were pots, ten chili seeds per pot, three pots for the replicas

Probably petri plates and it shouldn’t be experimental unit. Replace à petri plates containing inoculated seeds/ petri plates containing treated seeds were placed …………………

As this changed for clarity (line 191), we did not replace the text as suggested

IMPORTANT: Variety of chilli and source of collection of seeds?

Guajillo chili seeds were obtained from a commercial source with a guarantee of 92 % genetic homogeneity and 95 germination percentage

Line 183: This variable is also called germination capacity (redundant sentence)

The phrase has been deleted

Section 2.6: Polyphasic analysis of Trichoderma (BMH-0059) and Aspergillus (BMH-0060) strains à Polyphasic analysis of selected strains

This has been changed

Section 2.6: Need to rewrite in a scientific language. Poor write-up i.e line 234 we used ……

Thank you for your observation. Now it reads (lines 255-259 v2):

“For strain BMH-0059, concatenated alignments of the DNA sequences for RPB2, Elongation Factor 1, and ITS were used. For the strain BMH-0060 we used the DNA sequences of RPB2, calmodulin, β-tubulin, and ITS (both ITS regions were previously reported by [26]) were used.”

Definitely the authors performed this study but need significant improvements in this section.

Here we mention step by step the method to reconstruct the phylogenies, we understand it may look a bit arid, but that is the proper description (except for your suggestion in which we changed “we used”… for “were used”…at the end of the sentence.

Line 239: Other options were set as default. (Redundant)

The phrase was deleted

Section 2.7: Effect of inoculation on C. annuum greenhouse conditions (what was studied on C. annuum under greenhouse conditions. Need to rephrase the title accordingly. Probably, the author studied the effects o growth of C. annuumin under greenhouse conditions.

The title of this section has been changed to: “Effect of inoculation of two selected strains on the growth of C. annuum in greenhouse conditions”

In this section, re-write the treatments in following order. T1, T2, T3 ….. etc and then elaborate the description of treatments.

This comment is confusing, in the manuscript we do not use the nomenclature T1, T2, T3, etc. Each strain has only its laboratory collection name BMH-00X (Biología Molecular de Hongos -00X)

Something happened when we uploaded the manuscript in the JoF platform; for example, all the shaded boxes lost the shade; some dots became commas, etc. The manuscript that I downloaded from the JoF site, as indicated, had already these mistakes… (which I am sure there were not in the original submission).

This section has poor English i.e line 265 the plants were watered..!

English has been revised and several changes were made. For example, “…watered” has been changed to “irrigated”

Line 275: use the word growth parameters instead of biometric.

This has been corrected all through the manuscript

Section 2.8:  ¿?

Results:

Section 3.1: Which strains were isolated from the roots? Need to illustrates categorically.

This information is contained in Table 1. In Figure 2 we show the macro and micro morphology of the isolates

 Line 329: to grow together?, disc size ? how old the mycelial disc was?ç

We added a better description of how this experiment was performed (lines 166-170 of the revised manuscript):

The 31 fungal morphotypes were grown on PDA medium for 6-8 days (depending on the growth speed) and solutions adjusted at 7x10-6 mL-1 spores were prepared. From this solutions, one microliter of each suspension was sown in opposite sides of the dish for each combination which contained two, three, or four strains in the same Petri dish (Figure 1) and incubated for 5 days at 25 °C ± 1.

A representative illustration is shown in Figure 1

Anyway, we changed the word “together” for “…in the same dish”.

Line 345: tests in plants à in plants experiments/ in planta assay

Thank you for your suggestion, we have modified the text accordingly

Line 352: I would suggest elaborating the concentrations as high, medium, low, etc depending upon the concentration i.e Low (7x104), medium (7x106), High (7x108) etc that would be easy to understand by reader and to clarify the results differentiation.

We have added this in the materials and methods section: (lines 184-185 of the revised manuscript):

… at three different concentrations: 7×104 (low), 7×106, (medium) and 7×108 spores mL-1 (high) of each fungal strain.

We also changed the numbers by “low, medium or high”, in the text and table 2, accordingly

In tables, the author mentioned shaded boxes; however, I couldn’t find the colour shades.

We are sorry for this inconvenience; the original manuscript had shaded boxes. Something happened when uploading the manuscript. This has been again depicted

Line 377: we decided to further characterize themà further characterization ………were performed.

This has been changed (line 401 in the new revised version)

Line 377-384: These should be in methodology section not results.

We have deleted most of that text so now it reads: (lines 405-409 of the revised manuscript):

as described in material and methods. by using sequences from molecular markers for β-tubulin, calmodulin, Elongation Factor 1, and RNA polymerase II, which were amplified and sequenced. The phylogenetic reconstruction was performed with reference sequences and with the Maximum Likelihood algorithm.

Figure 3, 4: The phylogenetic tree is unclear (even after zoom out) this needs to be replaced with the original image:

The Figures have been enhanced

Line 413: kept increasing? à Constant increase till 45 days was recorded.

This has been changed as you suggest

Section 3.6: This section needs explanation with data among the treatments.

We have added the following paragraph (lines 441-448 of the revised version):

“In Andisol, foliar P concentration varied through time with no consistent pattern, independently of the inocula and was higher or at best equal without RP. The highest concentration of foliar P in this soil was at day 21 when inoculated with both strains and without RP. In Vertisol with RP, the highest amount of foliar P was achieved at day 15 when the plants were inoculated dually and did not change with time. In contrast, when vertisol was not added with RP, the best performance regarding foliar P increased with time and was higher when the plants were inoculated with strain BMH-0060 (the best solubilizer) or with the dual inoculation was applied.”

Figure 5,6 : Alphabet letters (level of significance) above the figures are missing.

They have been added in the figure

Line 461-463: Not the results

You are right, sorry. We deleted “(Figure 9)” [which is the PCA analysis) and stated “(data not shown)”

In section 3.7: The results and methodology are combined i.e. line 481-482; 487-492;

We think that this text is necessary to be friendlier with the reader, so we left it as it is to avoid having to back to the Materials and Methods Section

Line 504,505, 506: scientific names should be Italic.

This has been corrected

Discussion

Line 552: Confirms the findings of? ( it should be xyz et al)

It has been added: Zúñiga-Silgado et al., 2020 (although we are not sure if this is the right format asked by the Journal)

Line 560: Need to mention the concentrations of what? Trichoderma or Aspergillus

Here we refer to fungi in general. Very few works have correlated the number of spores with the growth of chili considering soil mineralogy

Line 562: Need to mention the author(s) of reported stud. Xyz et al

To be friendlier with the reader we changed the sentence to: “…by other groups [21, 43, 44]”

Line 564: Yes, the same should be throughout the manuscript (lower / higher concentration etc)

Yes, this has been considered

Line 567: Write down the full form of PSF and PMF for first time and later use abbreviation

We are sorry, now it is written: “…Phosphorous Solubilizing Fungi (PSF) and Phosphorous Mineralizing Fungi (PMF)

Line 572: as previously reported.

It has been corrected

Line 578-580: I think this is not the part of discussion; it might be a recommendation/ conclusion by other studies.

The sentence has been moved to the Conclusions section

Line 584-588: How you claim that this process was mass flow rather than diffusion?

We do not claim it is by mass flow only, the way the sentence is written does not discards diffusion. However, we have changed the sentence to “…However, in the case of poorly mobile nutrients such as P, only small amounts reach the roots by mass flow, although some diffusion could also be possible.”

Line 589-590: The author claimed we expected but the references describe the above statement..!

It is not really like that, [16] does report the effect of Trichoderma on P solubilization but does not consider the soil type; the other cited reports use other microorganisms (Bacillus, for example or other Trichoderma spp.) and other types of soils, so the references are just to stress that soil properties affect P uptake.

Line 619-620: The allosteric….. these nutrients (unclear)

We are sorry, this phrase was carried along from a previous version of the manuscript. It has been deleted.

Line 620-621: manuscript in preparation (this is not the justification of present study)

This has been changed by “(data not shown)”

Line 628: the solution was found……. (This study is previously reported) what about the results of present study in discussion section?

In this study we did not measure P in solution, we only did it in foliar tissue.

Line 629-633: Justify with xyz et al and then mention reference. Our results are in the agreement with xyz et al [00].

This has been corrected

Line 633-634: negatively affected root growth and inhibited seed germination in different plants à negatively affects the growth of roots and ultimately hindered the seed germination of different plants.

Thank you for your suggestion, it has been considered

Line 634-638: too long sentences

The sentence has been modified: “According to our results, the inhibition of the germination and growth of chili seedlings in Andisol when inoculated with T. virens and/or A. tubingensis could be explained by an increase in rhizospheric acidity combined with the physical-chemical properties inherent to the soil type. This significantly detracts the growth variables evaluated [56]”

Conclusion is not sufficient, need a comprehensive conclusion indication the importance of present study in term of chilli agricultural production. Further, if possible what kind of recommendations/suggestions by the author (s) are !

The conclusion section has been modified accordingly

Reviewer 2 Report

Topic is divided in discontinues sections. Revise the title to make the meaning clear 

Abstract should contain some numerical data. Abstract should be started with a concise introduction or background of your title or objectives not a general introduction of only plant

Words already present in title should be avoided to add in keywords

Second paragraph should be merged with first one

I did not find the repetition in green house experiment

 3.6. Greenhouse evaluation of chili plants..... meaning is not clear.... did you evaluate chili plants  ?

No test was applied to data in Figure 5 and 6 to know the significant difference among different treatments

Conclusion is too short. Conclude the whole findings in this section 

Author Response

Answers to Reviewer 2:

We want to thank you for your comments, they will undoubtedly make our work better

Topic is divided in discontinues sections.

This is the format required by the Journal (1. Introduction, 2: Materials and Methods, which can be subdivided in specific methods (2.1; 2.2; 2.3, etc.); Results, which can also be divided in different groups of experiments according to the topic, etc.

Revise the title to make the meaning clear 

The title has been modified

Abstract should contain some numerical data. Abstract should be started with a concise introduction or background of your title or objectives not a general introduction of only plant

The abstract has been modified as requested by the reviewer

Words already present in title should be avoided to add in keywords

The Key words have been modified

Second paragraph should be merged with first one

Done

I did not find the repetition in green house experiment

This was presented as supplementary material, now it is a Figure is included in the main text

 3.6. Greenhouse evaluation of chili plants..... meaning is not clear.... did you evaluate chili plants  ?

We have changed the subtitle, now it reads “3.6. Greenhouse evaluation of chili plant performance in different kinds of soils”

No test was applied to data in Figure 5 and 6 to know the significant difference among different treatments

Figures 7 and 8 show the statistical tests of the data presented in Figures 5 and 6. (so the figures would not be too “crowded”). Nevertheless, we have modified figures 5 and 6 which now show above the bars the ANOVA significance values

Conclusion is too short. Conclude the whole findings in this section 

The Conclusion section has been modified accordingly.

Reviewer 3 Report

Comments to the author JoF (1986478)

The manuscript entitled ““Milpa” isolates of Trichoderma virens and Aspergillus tubingensis promote solubilization, mineralization and phyto absorption of phosphorus in Capsicum annuum L depending on soil type” is really (really.!) interesting one and the author (s) performed a nice work. I truly enjoyed the reading of the manuscript. The study by Zúñiga-Silgado et al. demonstrated the use of Trichoderma and Aspergillus as promoter for capsicum. Overall, the manuscript is well structured; presenting novelty and authenticity of work. The results are reliable and manuscript is in accordance with the Journal’s scope. However for the improvements, major revision is compulsory before further consideration.

In abstract, the author mentioned the methodology; it should not be the part of abstract, for example:

Line 37-40,

There are no results in abstract only the line 41-43

Further, in abstract, the conclusion of work must be concise and possible future aspects should be illustrated.

Introduction

Line 49: and
à which belongs

Line 55-56 à dollar $

Line 56-57 repeating sentence in abstract and intro

Line 59-60: “thus is one of the most critical factors limiting plant growth” what does it means which factor?

Line 76-77: repeating sentences with line 63-64

Line 77: agronomic management or Agronomic practices?

Line 79: satisfy à use appropriate word

Line 81-85: Too long sentence

Line 86: changes in the soil (what kind of changes?)

Line 92-93: “Microbial consortia have been used successfully as biofertilizers in various crops of agronomic importance in Mexico such is the case of chili” à Microbial consortia as biofertilizers have been successfully used in various crops of agronomic importance including chili in Mexico.

Line 98-100: Unclear sentence, need to rephrase

Line 104-107: long sentence, need appropriate rephrasing

General comments for introduction:  

Insufficient information for the utilization of Trichoderma and Aspergillus as biofertilizer and/or as plant growth promoter.

Materials and methods

Section 2.2. No information regarding the incubation of strains, either fungal or bacterial. As the author mentioned PDA, I believe that for the isolation different fungal and bacterial microorganisms are present as these are 300 colonies. Thus, need to clearly elaborate this section. 

Section 2.4: the author tested thirty one isolates, but later on only four strains were mentioned (line 149). There is a confusion that how these strains were selected and either these strains were among the thirty one?

Section 2.5: How the seeds were inoculated, by spraying or seed priming method? And he most important, how long (time duration) the seed inoculation was?

How the spore suspension was prepared and how the spores’ suspension was adjusted to working concentrations?

What is experimental unit? Containing? Petri plates/pots ?

Probably petri plates and it shouldn’t be experimental unit. Replace à petri plates containing inoculated seeds/ petri plates containing treated seeds were placed …………………

IMPORTANT: Variety of chilli and source of collection of seeds?

Line 183: This variable is also called germination capacity (redundant sentence)

Section 2.6: Polyphasic analysis of Trichoderma (BMH-0059) and Aspergillus (BMH-0060) strains à Polyphasic analysis of selected strains

Section 2.6: Need to rewrite in a scientific language. Poor write-up i.e line 234 we used ……

Definitely the authors performed this study but need significant improvements in this section.

Line 239: Other options were set as default. (Redundant)

Section 2.7: Effect of inoculation on C. annuumin greenhouse conditions (what was studied on C. annuumin under greenhouse conditions. Need to rephrase the title accordingly. Probably, the author studied the effects o growth of C. annuumin under greenhouse conditions.

In this section, re-write the treatments in following order. T1, T2, T3 ….. etc and then elaborate the description of treatments.

This section has poor English i.e line 265 the plants were watered..!

Line 275: use the word growth parameters instead of biometric.

Section 2.8:

Results:

Section 3.1: Which strains were isolated from the roots? Need to illustrates categorically.

 Line 329: to grow together?, disc size ? how old the mycelial disc was?

Line 345: tests in plants à in plants experiments/ in planta assay

Line 352: I would suggest elaborating the concentrations as high, medium, low, etc depending upon the concentration i.e Low (7x104), medium (7x106), High (7x108) etc that would be easy to understand by reader and to clarify the results differentiation.

In tables, the author mentioned shaded boxes however; I couldn’t find the color shades.

Line 377: we decided to further characterize themà further characterization ………were performed.

Line 377-384: These should be in methodology section not results.

Figure 3, 4: The phylogenetic tree is unclear (even after zoom out) this needs to be replaced with the original image.

Line 413: kept increasing? à Constant increase till 45 days was recorded.

Section 3.6: This section needs explanation with data among the treatments.

Figure 5,6 : Alphabet letters (level of significance) above the figures are missing.

Line 461-463: Not the results

In section 3.7: The results and methodology are combined i.e. line 481-482; 487-492;

Line 504,505, 506: scientific names should be Italic.

Discussion

Line 552: Confirms the findings of? ( it should be xyz et al)

Line 560: Need to mention the concentrations of what? Trichoderma or Aspergillus

Line 562: Need to mention the author(s) of reported stud. Xyz et al

Line 564: Yes, the same should be throughout the manuscript (lower / higher concentration etc)

Line 567: Write down the full form of PSF and PMF for first time and later use abbreviation

Line 572: as previously reported.

Line 578-580: I think this is not the part of discussion; it might be a recommendation/ conclusion by other studies.

Line 584-588: How you claim that this process was mass flow rather than diffusion?

Line 589-590: The author claimed we expected but the references describe the above statement..!

Line 619-620: The allosteric….. these nutrients (unclear)

Line 620-621: manuscript in preparation (this is not the justification of present study)

Line 628: the solution was found……. (This study is previously reported) what about the results of present study in discussion section?

Line 629-633: Justify with xyz et al and then mention reference. Our results are in the agreement with xyz et al [00].

Line 633-634: negatively affected root growth and inhibited seed germination in different plants à negatively affects the growth of roots and ultimately hindered the seed germination of different plants.

Line 634-638: too long sentence

Conclusion is not sufficient, need a comprehensive conclusion indication the importance of present study in term of chilli agricultural production. Further, if possible what kind of recommendations/suggestions by the author (s) are !

Author Response

(The authors gave the same response as above.)

Round 2

Reviewer 3 Report

The manuscript by Zúñiga-Silgado et al. has been significantly improved. However for the improvements, some minor points are needed to be covered before further consideration.

Line 36-37: Move these lines into introduction, it’s totally redundant here

I feel that abstract should be re-structured containing following sequence (Short introduction and background of C. annuum, then solubilization, mineralization and phytoabsorption of phosphorus, aims of study in a short way, results with explanations using the numerical data of best treatment, suggestion and conclusion).
No need to mention about previous work in abstract (need to focus only on present study)

I still feel that English should be improved by appropriate person, expert in this field.

For example: Daily observations were made for 15 days, during which the number of germinated seeds was recorded à Experiment was monitored for fifteen days and seed germination was observed. Seeds were said to germinate if the length of radical was 2mm. Germination percentage was calculated from the recorded data and germination speed index (GSI) was measured using a reported formula…………….

The same revision needed for some parts of the manuscript.

Source of chili seed collection and any variety name ?

Line 287: 80 experimental units or 8? Because from treatments I feel it should be 8, confirm please.

Line 290: why deionized water? And field capacity ? but it is pot experiment.

Line 293: Need to mention the age of plants, how old plants were sampled?

Line 294: dissected àexcised

Line 296: developer solution ?

Line 298: at 298nm à λ = 298nm

As mentioned, some sections needed a minor revision for the accuracy and improvement of manuscript.

For example section 2.8: A complete randomized design with a 4x4 factorial arrangement (4 strains and 3 doses of inoculum evaluated and 1 uninoculated control) for the in vitro chili seed germination was used. Three replicates were used for each treatment whereas (3/4/5/6) Plates were subjected as a replicate.

No need to mention the dependent variable etc.

Line 318: the obtained data were analyzed with a Principal Component Analysis (PCA) and performing ANOVA. (Need to mentioned the ANOVA either one way-two way etc )

Author Response

The manuscript by Zúñiga-Silgado et al. has been significantly improved. However, for the improvements, some minor points are needed to be covered before further consideration.

Line 36-37: Move these lines into introduction, it’s totally redundant here

In the downloaded document from the JoF editorial system the lines that you mention state:

“In 2021 Mexico produced 3.3 billion tons encompassing 45, 000 hectares which yielded 2 billion dollars in exports to the USA, Canada, Japan, etc. The aim of this work was to study the effects of…”

I see no redundancy in this context, so I assume you refer to lines 33-34 which could be redundant with the previous sentence:

“A significant effect regarding bioavailable P release was observed in the presence of the fungi.”

This line has been deleted, since in the introduction section would be redundant too

I feel that abstract should be re-structured containing following sequence (Short introduction and background of C. annuum, then solubilization, mineralization and phytoabsorption of phosphorus, aims of study in a short way, results with explanations using the numerical data of best treatment, suggestion and conclusion).

We have re-structured the abstract accordingly. However, we believe that inserting all the numerical data would make the abstract too long and difficult to interpret without a context. This is given in the Results Section. However, we mention the results, suggestion, and conclusion of the work. We only show numerically the results of the best treatment which was dual inoculum with 7 x 106 spores mL-1.

No need to mention about previous work in abstract (need to focus only on present study)

If we do not mention our previous work, then lines 33-34 are not redundant and necessary to give context to the study. And in fact, this is the follow up of the previous study which is described with some more detail in the introduction Section. So, we decided to leave that sentence as it is and deleted lines 33-34 following your suggestion.

I still feel that English should be improved by appropriate person, expert in this field.

For example: Daily observations were made for 15 days, during which the number of germinated seeds was recorded à Experiment was monitored for fifteen days and seed germination was observed. Seeds were said to germinate if the length of radical was 2mm. Germination percentage was calculated from the recorded data and germination speed index (GSI) was measured using a reported formula…………….

The same revision needed for some parts of the manuscript.

It is surprising that you mention this, since two of the co-authors are native English speakers (Dr. Jeffrey J. Coleman and Miranda Otero). They have read the manuscript thoroughly at least three times. Dr. Coleman is an Associate Professor & Graduate Program Officer at Auburn University and has 4,163 citations to his works.

The corresponding author has more than 75 publications in JCR indexed Journals with more than 1,700 citations. So, we think English issues are only a matter of style, the meaning is clear as it is written, which is important for the reader.

Source of chili seed collection and any variety name ?

In lines 196-197 the following sentence has been added:

“Guajillo chili seeds were obtained from a commercial source with a guarantee of 92 % genetic homogeneity and 95 germination percentage. They were surface disinfected…”

We are not allowed to disclose the name provider for commercial publicity reasons in Mexico

Line 287: 80 experimental units or 8? Because from treatments I feel it should be 8, confirm please.

The correct number is 80. A lot of work! Here are the numbers (which are clearly described just below that line: 4 inocula (BMH-0059, BMH-0060, BMH-0059 PLUS BMH-0060 and one control without inocula, so here we have 4. Two P conditions: with or without RP [4 x 2=8] x 2 types of soils [4 x 2 x 2 = 16] x 5 replicas (pots per treatment) [4 x 2 x 2 x 5 = 80]

Line 290: why deionized water? And field capacity? but it is pot experiment.

The plants were irrigated alternatively with deionized water and Hoagland medium as described in lines 293-294 to avoid salinization of the soil, which obviously would affect the experimental results.

“Field capacity” is an agronomical term that measures the capacity of a given soil to retain water, it has nothing to do whether it is in a field or in a pot. If the plants were much below field capacity, they would suffer from drought stress, if the level of humidity is way above field capacity the plants would also suffer stress.

Line 293: Need to mention the age of plants, how old plants were sampled?

This is clearly stated in lines 293-296:

“The plants were irrigated daily with deionized water up to 60% field capacity. Every 8 days the plants were irrigated with a P-free Hoagland solution [35, 36], and the foliar P concentration was determined. At four different time points (15, 21, 37, and 45 days post-inoculation), one leaf disc of 0.5 cm Ø from the youngest leaf…”

For the biomass experiments the age of the plants is clearly described in lines 302-303:

“The experiment lasted 45 days, after which all treatments were disassembled. The following growth parameters were evaluated in each experimental unit…!

Line 294: dissected à excised

Done

Line 296: developer solution?

The reference of the method (which includes the formula of the developer solution) is given to avoid unnecessary text. For example, we did not describe the composition of the Hoagland solution, this is found in the references to the original methods. We only mentioned solution compositions when were modified from the original reference (for example lines 141-146).

Line 298: at 298nm à λ = 298nm

The method to measure P is performed at 890 nm, not 298. And when we mention a spectrophotometer and the units (nm) implies we are measuring wavelength. Anyway, we have added l as suggested.

As mentioned, some sections needed a minor revision for the accuracy and improvement of manuscript.

For example, section 2.8: A complete randomized design with a 4x4 factorial arrangement (4 strains and 3 doses of inoculum evaluated and 1 uninoculated control) for the in vitro chili seed germination was used. Three replicates were used for each treatment whereas (3/4/5/6) Plates were subjected as a replicate.

No need to mention the dependent variable etc.

We believe that it is clearer the way it is written, especially for non-specialists or post-graduate students. With all due respect even you got confused with the experimental units and treatments in line 287, where it is clearly stated which are the variables and the experimental unit.

Line 318: the obtained data were analyzed with a Principal Component Analysis (PCA) and performing ANOVA. (Need to mentioned the ANOVA either one way-two way etc )

We used a Two-way ANOVA. Now this has been added to the text.